# Discovery of flat seismic reflections in the mantle beneath the young Juan de Fuca Plate

Yanfang Qin [1,2], Satish C. Singh [1✉], Ingo Grevemeyer [3], Milena Marjanović[1] & W. Roger Buck[4]

Crustal properties of young oceanic lithosphere have been examined extensively, but the nature of the mantle lithosphere underneath remains elusive. Using a novel wide-angle seismic imaging technique, here we show the presence of two sub-horizontal reflections at ~11 and ~14.5 km below the seafloor over the 0.51–2.67 Ma old Juan de Fuca Plate. We find that the observed reflectors originate from 300–600-m-thick layers, with an ~7–8% drop in P-wave velocity. They could be explained either by the presence of partially molten sills or frozen gabbroic sills. If partially molten, the shallower sill would define the base of a thin lithosphere with the constant thickness (11 km), requiring the presence of a mantle thermal anomaly extending up to 2.67 Ma. In contrast, if these reflections were frozen melt sills, they would imply the presence of thick young oceanic lithosphere (20–25 km), and extremely heterogeneous upper mantle.

[1] Institut de Physique de Globe de Paris, 1 rue Jussieu, 75238 Paris, France. [2] Now at Japan Agency for Marine-Earth Science and Technology (JAMSTEC), Showa-machi 3173-25, Kanazawa-ku, Yokohama 236-0001, Japan. [3] GEOMAR, Helmholtz Centre for Ocean Research Kiel, Wischhofstr 1-3, 24148 Kiel, Germany. [4] Lamont-Doherty Earth Observatory, Columbia University, 61 Route 9W, Palisades, NY 10964-1000, USA. ✉email: singh@ipgp.fr

The plate tectonic theory requires the presence of a rigid lithosphere floating over a ductile asthenosphere. In the oceanic domain, the plate-cooling model suggests that the lithosphere should be thin near the ridge axis and thicken as a plate moves away from the ridge axis[1]. However, as the crustal accretion[2] and active hydrothermal circulation processes occurring near the ridge axis modify the thermal regime significantly[3,4], the thickness of the lithosphere of a young oceanic plate should be higher than that predicted by the plate cooling model[1].

The thickness of the lithosphere has been traditionally estimated using surface wave studies[5–8] but as long period (>20 s) surface waves with wavelengths >80–100 km[9] are generally used, imaging the base of a young, thin lithosphere has proven challenging[9].

The lithosphere–asthenosphere boundary (LAB) has also been imaged using receiver function methods, indicating a sharp S-wave velocity contrast at the base of the lithosphere[10–12]. A single ocean-bottom seismometer (OBS) receiver function study near the Juan de Fuca Ridge indicated the presence of an 8–15-km-thick high-velocity mantle over a low-velocity layer, suggesting that lithosphere beneath the OBS could be 14–21-km thick[13]. However, in order explain the receiver function signal, a 26–30% of anisotropy in 8–15-km-thick layer was required, requiring unacceptably large velocity variations (e.g. a P-wave velocity $8.0 \pm 1.2\ \mathrm{km\,s^{-1}}$) below the Moho. Another receiver function study covering 0–8 Ma of the Juan de Fuca (JdF) Plate provided a phase converted S-wave sub-horizontal image at ~30-km depth beneath the sea surface. However, this receiver function image spans in a wide depth range (±10 km)[14], indicating that the receiver function imaging methods have a limited resolution to precisely decipher the thickness of an young lithosphere.

Recently, seismic reflection methods have been used to image the LAB for the old oceanic lithosphere with much more precision[15,16], indicating that the LAB is also associated with a sharp P-wave velocity contrast. These results indicated two reflections, associated with the top and bottom of the LAB, arguing that the LAB represents a melt channel at the base of the oceanic lithosphere. A magnetotelluric study conducted in the Middle America Trench imaged a thin, high conductivity anomaly, which is attributed to the presence of a partial melt channel, corroborating the results of the controlled source seismic studies[17]. In addition, a wide-angle seismic method revealed mid-lithospheric discontinuities between 37 and 59 km depth over the 128–148 Ma old Pacific Plate, which are interpreted as frozen melt sills[18]. However, the nature of the LAB near the ridge axis for a young oceanic plate remains largely unidentified. Here, we present the P-wave image of a young (0.51–2.67 Ma) oceanic lithosphere using a novel approach of wide-angle seismic reflection imaging.

Our study area lies on the JdF Plate, covering the 0.51–2.67 Ma old oceanic crust formed along the Endeavour segment of the intermediate-spreading JdF Ridge (Fig. 1). This ~90-km-long ridge segment is bounded by the Endeavour–West Valley overlapping spreading centre (OSC) in the north and the Cobb OSC in the south. On the western flank, this segment is characterized by the prominent Heckle seamount chain (Fig. 1a), resulting from the presence of a small-scale mantle thermal anomaly and the north-westward advance of the JdF Ridge (30 mm yr⁻¹) in the fixed hotspot reference frame[19–21]. An enhanced crustal production since 0.71 Myr and the presence of an ~40-km wide and 300-m high plateau are linked to this mantle thermal anomaly[22,23]. In contrast to the seamount dominated west flank, the seafloor on the east flank is flat and covered with thick sediments[24,25].

The tectonic history of the Endeavour segment is dominated by several episodes of ridge propagation starting from the northward propagation of the Cobb OSC at ~4.5 Myr ago[26]. Coincident with

the onset of the activity of the Heckle melt anomaly ~2 Myr, the Cobb OSC propagated southward for about 35 km[19]. For young crustal ages (<200 ka), a small eastward jump of the Endeavour segment has been indicated due to the readjustment of the plate boundary, and the influence of the Heckle melt anomaly[22]. Finally, numerical modelling of the tectonic fabric suggested that the northward propagation of the Cobb OSC has been restored in the last 100 ka[27]. This change in the direction of the ridge propagation could be associated with the diminished melt supply to the ridge axis due to the transfer of the Heckle melt anomaly to the east flank[22]. Recently, active-source seismic studies showed that there is a lateral offset between the mantle and the crustal magmatic systems[28,29] (Fig. 1b), which is attributed to the difference in mechanisms of heat transfer that operate in the crust (advection) and mantle (conduction)[29]. Interestingly, the most prominent low-velocity zones lie directly beneath the OSCs. In addition to the tectono-magmatic processes, the Endeavour segment is highly influenced by hydrothermal processes. The rift valley in the central part of the Endeavour segment hosts five long-lived vent fields underlain by the existence of intra-crustal melt lenses at ~2.6 km below the seafloor[22].

Four ridge-parallel wide-angle seismic profiles (RFR96-01, -03, -05, and -08) were acquired aboard R/V Sonne in 1996 (Fig. 1b). Along each of the profiles, six ocean-bottom hydrophones (OBHs) were deployed at 4 km interval. The details of the acquisition parameters are given in the "Methods" section. The OBH data contain strong crustal arrivals (Pg) and reflections from the Mohorovičić discontinuity—Moho (PmP). In addition, we observe wide-angle reflections that seem to originate in the mantle at a source-receiver offset range of 7–20 km, ~1–2 s after Pg arrivals (Fig. 2 and Supplementary Fig. 1).

We apply an advanced imaging technique to these wide-angle reflection data. We find two sub-horizontal reflections at ~11 and 14.5 km below the seafloor in the mantle over 0.5–2.67 Ma JdF Plate and suggest that they could either represent the LAB or frozen melt sills in the lithospheric mantle, requiring the presence of steady-state melt sills beneath the ridge axis.

## Results

**Velocity model**. For constraining the crustal thickness and velocity structure sampled by the profiles, we first performed ray-based travel time tomography of Pg and PmP arrivals (Supplementary Fig. 2; see the "Methods" section). The observed crustal thickness is consistent with that observed along the orthogonal seismic reflection profile[22,30]. In the absence of any velocity constraints from the mantle turning ray arrivals (Pn), we used a gradually increasing age (temperature) dependent one-dimensional velocity function below the Moho. The velocity just below the Moho varied from 7.65 to $7.8\ \mathrm{km\,s^{-1}}$ for the youngest to the oldest profile, respectively (Supplementary Fig. 3). We then employed travel time tomography to estimate the depth of the mantle reflections (Supplementary Fig. 3). A synthetic seismogram modelling tests (Methods section) indicates that these arrivals are not multiples or artefacts, but real reflection arrivals from the mantle (Supplementary Fig. 4).

**Wide-angle seismic image**. To obtain a seismic reflection equivalent image, we performed a pre-stack depth migration[31,32] of the reflected part of the wide-angle seismic data by using the above velocity models (Methods section; Supplementary Fig. 3). Prior to the migration the data were downward continued near the seafloor (only the source-side), so that both sources and receivers are at approximately the same depth. A wave equation datuming method was used for the downward continuation[33,34]. To avoid migration artefacts, the Pg and P-to-S converted waves,

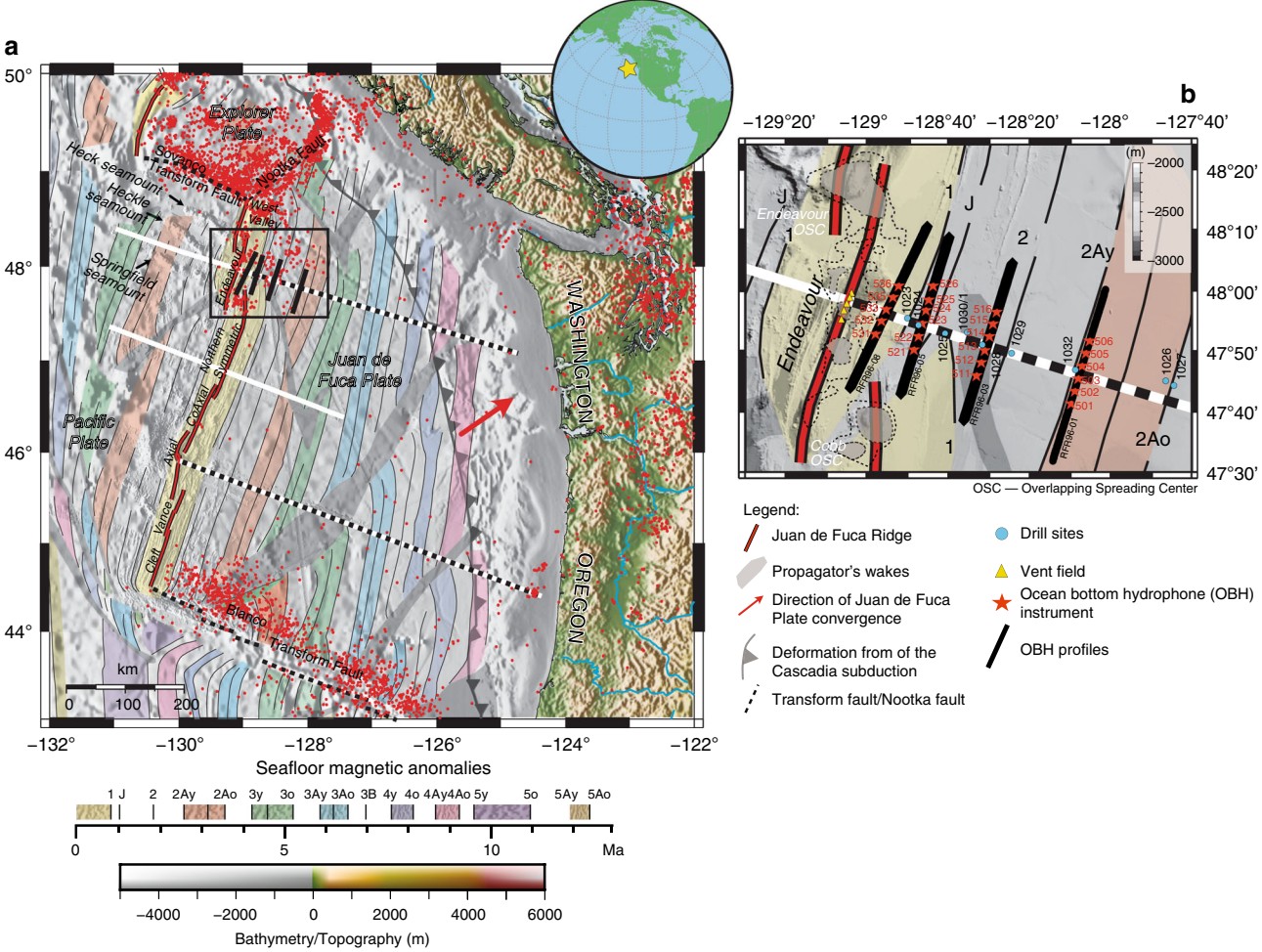

**Fig. 1 Study area and the young Juan de Fuca (JdF) Plate. a** Regional map of the JdF Plate and Cascadia subduction zone. The inset globe shows the global location of the area marked with the yellow star. Our study area is outlined by a black box (close-up is shown in panel **b**). The locations of the seismic reflection profiles collected in 2002 (refs. [21,53]) crossing the Endeavour and Northern Symmetric segments are shown in white lines; the locations of the reflection seismic profiles collected in 2012[41] are shown in white-black lines. Coloured areas bounded by black lines represent the magnetic anomalies with the corresponding age shown in the time scale below the panel[53]. Grey shaded zones outline the extent of the propagator wakes[53]. The red dots represent epicentres along plate boundaries (June 1995–June 2019, United States Advanced National Seismic System and Canadian National Seismic Network catalogues). **b** The close-up of the study area. Around the ridge axis, dashed and grey transparent contours outline zones with lower (~0%) and higher (3%) melt fraction at 7.8-km depth[28]. The remaining symbols are shown in Legend.

and water bottom multiples were muted. A Kirchhoff pre-stack depth migration technique[31,32] was applied to the resulting downward continued OBH data to map the seismic reflection events to the appropriate depth and distance locations in the sub-surface. The velocity model obtained by the tomography for profile RFR96-01 was used to compute the Green's function for the migration. The final migrated gathers were summed, and the seismic images are displayed in Fig. 3 and Supplementary Fig. 5. In order to make sure that our pre-stack depth migration procedure is accurate, we performed a pre-stack depth migration of the synthetic data after performing all the pre-processing steps that were applied to the real data. The final image shows that the synthetic pre-stack depth migrated image contains Moho and two mantle reflections accurately (Supplementary Fig. 6). To verify that the migrated images are consistent with reflection arrivals, we also present the post-stack time migrated image (Supplementary Fig. 7). The similarities between the pre-stack and post-stack migrated images confirm that they are real images originating from the mantle. The depth obtained using travel time modelling of these reflections is consistent with the migrated images (Supplementary Fig. 8).

Uncertainties in the depth of these reflections are primarily due to the uncertainties in the mantle velocity structure. Supplementary Fig. 8 indicates that the upper reflection lies at $11 \pm 0.5$ km and the lower reflection at $14.5 \pm 0.5$ km. To assess uncertainty in the velocity model, we performed a pre-stack depth migration of OBH data along profile RFR96-01 using three different velocity models: (1) lower limit of the tomography velocity model, (2) the tomographic velocity model and (3) upper limit of the tomographic velocity model. Supplementary Fig. 9 shows that the image obtained using the lower velocity model is similar to that with the tomographic velocity model, but the higher velocity model produces a lower quality image. The best image is obtained using the tomographic velocity model, suggesting that the velocity model used for the migration is satisfactory, and the image is a truthful representation of the sub-surface.

It should be noted that as the downward continued data were muted to remove the crustal turning rays, i.e., Pg and converted S-wave energy (Supplementary Fig. 7), the first 8 km of the migrated signals are reverberations and are not interpreted.

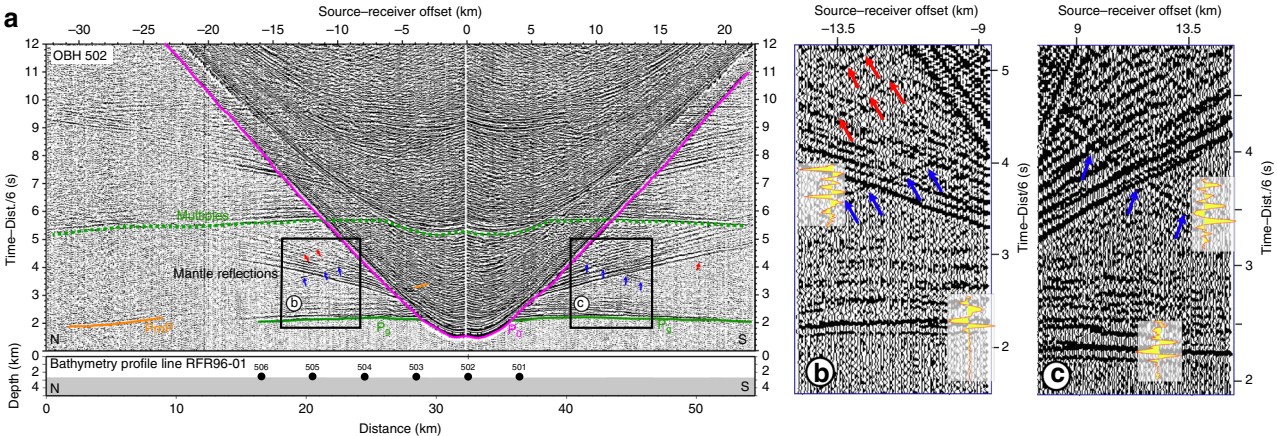

**Fig. 2 Wide-angle seismic data recorded on OBH 502 along profile RFR96-01. a** The primary seismic phases are direct water wave (P$_O$, magenta), crustal P-wave refractions (Pg, green) and crust-mantle boundary (Moho) P-wave reflections (PmP, orange). The presence of multiples is shown in a dashed green line. The mantle reflection events, the focus of this study, are marked in blue and red arrows. In green arrows, we mark P-to-S converted energy. The black boxes (**b**) and (**c**) show the close-up regions on the right focusing on the mantle reflection events (blue and red arrows). Yellow filled red curves indicate seismic waveforms for Pg and mantle reflected waves demonstrating the polarities of the mantle reflections. The bottom panel shows seafloor bathymetry along the line with the location of OBH indicated.

**Deep mantle reflections.** The depth migrated seismic images consistently show a bright reflection at an average 10.5 km below the seafloor for line RFR96-01 over the 2.67 Ma old crust (Fig. 3 and Supplementary Fig. 5). The other profiles show similar features, with small depth variations. For profile RFR96-05, the top reflection varies from about 10.2 to 11.5 km from north to south, while in the other two profiles, this event occurs at around 10.5 km. Overall, the top reflection is observed along all profiles; it can be followed for >20–25 km along each line and for ~65 km across the ridge axis approximately at a depth of 11 ± 0.5 km below the seafloor (Supplementary Fig. 8).

At about 3.5 km below this first reflector, i.e., at 14.5 ± 0.5 km below the seafloor, a second reflection can be observed along the older profiles for the 1.65 and 2.67 Ma old lithosphere (RFR96-01, RFR96-03, respectively). One could also see small segments of the deeper reflections in profiles RFR96-05 and RFR96-08 that sample ~0.9 and 0.51 Ma old lithosphere, respectively (Fig. 3a, Supplementary Figs. 5 and 7). In addition, there are several small reflections in between these reflections that can be identified in all of the profiles.

**Nature of the mantle reflections.** The quantitative nature of these mantle reflections could be obtained using a combination of (a) polarity analysis of the seismic signal, (b) synthetic seismogram modelling and (c) amplitude versus offset analysis. Although the low signal-to-noise ratio present in the data and the interference of different arrivals, the signal of the top reflection shows a reversed polarity with respect to the Pg arrivals (Fig. 2a) at certain locations, suggesting that it could be produced by a decrease in velocity at an interface.

The observed wide-angle reflections are limited in the 7 and 20 km source-receiver offset range. As we do not observe any large amplitude critical angle reflections, these reflections cannot originate from thick high-velocity layers. However, they could be produced either by thin or thick, low-velocity layers or thin, high-velocity layers. To assess the different possibilities, we carried out synthetic seismogram modelling for a series of one-dimensional models by varying both thickness and velocity of a layer embedded in the mantle peridotite. Previous studies indicated that the mantle reflections are produced by a 6–8.5% change in the P-wave velocity[15–18,35,36], and therefore, we used ±7% as an

average value. We found that an increase in velocity of +7% in a 1200-m-thick layer would produce two reflection arrivals, one from the top and the other from the bottom of each layer. The pair of top and bottom reflectors is not observed in the data (Supplementary Fig. 10). However, a 300-m-thick high velocity does produce reflections that resemble the observed data, but its polarity is positive (Fig. 4F). We then tested a −7% decrease in velocity, where the thickness of the layers varied from 300 to 1200 m (Supplementary Fig. 11). For the layers thicker than 600 m, there would be two reflection arrivals originating from the top and the bottom of the layer, suggesting that the layer thickness should be around 300 m. Finally, we compared the modelling results obtained by varying the velocity in a 300-m-thick layer by −7% (Fig. 4c and Supplementary Fig. 11c), −15% (Fig. 4d and Supplementary Fig. 12a) and −30% (Fig. 4e and Supplementary Fig. 12b). A −7% decrease in velocity could either correspond to the presence of a frozen gabbroic sill[18] or partially molten sill[16] within mantle peridotite, whereas a −30% decrease could be due to the presence of a large amount of melt[37,38]. We find that a decrease in velocity of −30% in a 300-m-thick layer would produce a strong reflection that is not observed in our data. We also tested the presence of a velocity gradient layer, where the velocity first decreases and then increases in a 600-m-thick layer, and the results are shown in Supplementary Fig. 13. The observed reversed polarity of the event, combined with the modelling results, lead us to suggest that the first mantle reflection is related to a negative velocity contrast, or a negative velocity gradient within a thin zone (<1/4 of wavelength). Similarly, the deeper reflections require the presence of a thin, low-velocity layer. Four different models and their associated synthetic seismograms are shown in Fig. 4.

A relatively poor signal-to-noise ratio does not permit to carry out thorough amplitude versus offset analysis, but we did compute the relative amplitude versus offset variations for the upper reflection (Fig. 5) and those for the best-fitting synthetic models (Fig. 4). The relative amplitude decreases very slowly with offset (from −1 to −0.95) for a velocity decrease of −30% in a 300-m-thick layer in the 8.5–17 km offset range whereas that more rapidly for −7% (from −1 to −0.65). An increase in the velocity of +7% produces a similar relative amplitude with offset, but it varies from +1 to +0.6 (i.e., the polarity does not match the observations), and therefore we can rule out an increase in

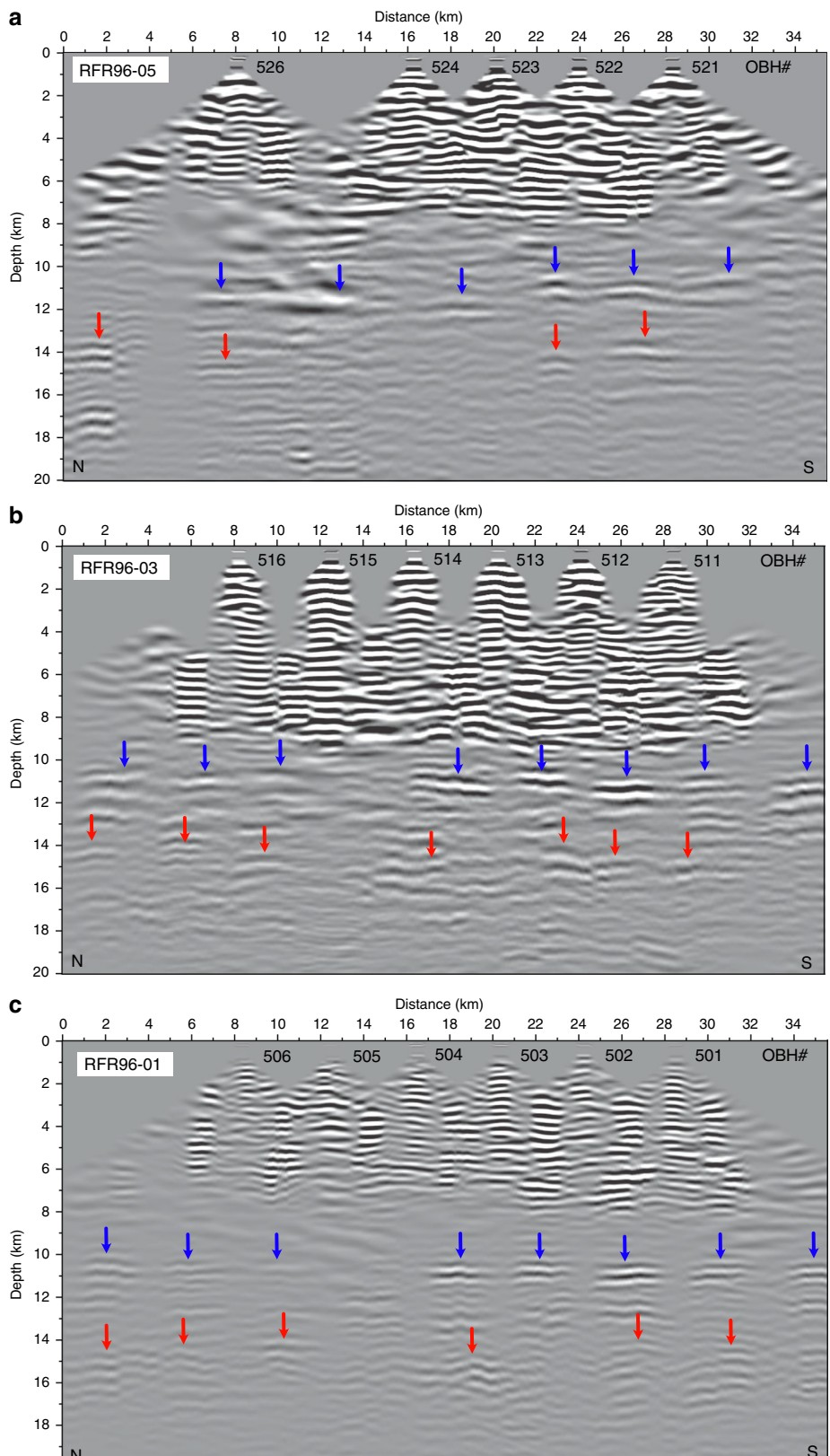

**Fig. 3 Pre-stack depth migrated images of the OBH profiles.** Prominent mantle reflections are indicated in blue (shallower) and red (deeper) arrows. **a** Profile RFR96-05 sampling crustal age ~0.9 Ma is shown. **b** Profile RFR96-03, sampling crustal age of ~1.65 Ma on average (please note that the line is crossing a propagator's wake spanning crustal ages from 0.78 to 1.86 Ma). **c** Profile RFR96-01 sampling almost uniform crustal age ~2.67 Ma. The blue and red arrows mark the locations of the two reflectors. The OBH instruments, numbered (Fig. 1), are indicated at the top of each panel. For profile RFR96-08 that samples ~0.51 Ma old crust, weak and interrupted events in the mantle are observed (Supplementary Fig. 5).

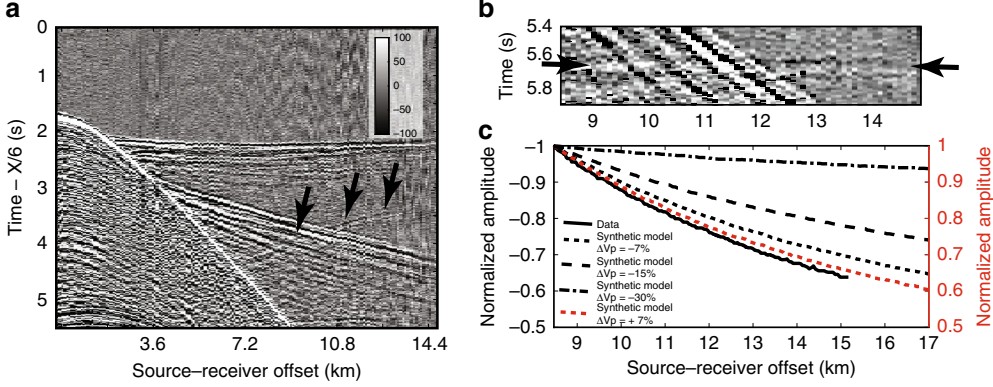

**Fig. 4 Synthetic seismograms. a** The observed data recorded by the instrument OBH 501, **b** a template for 1D velocity model where the upper mantle layer is 300-m thick with a velocity decrease of −7% and a 1200-m-thick lower layer with an increase in velocity of +7%. The black rectangle indicates the blow of the velocities shown in subsequent panels. **c** Synthetic seismograms for the velocity model shown in (**b**). **d** Synthetic seismogram for velocity in the upper layer of −15%, **e** −30% and **f** +7%.

**Fig. 5 Relative amplitude versus offset variation. a** Observed OBH 501 gather with a reduced velocity of 6 km/s highlighting the part of reflection event (black arrows) used for the amplitude versus offset (AVO) analyses. **b** Normal moveout corrected gather used for the AVO study. Black arrows highlight the mantle reflection. **c** Relative amplitude versus offset curves for real data (thick black) and synthetic data with different velocities (−7, −15, −30, +7%) in a 300-m-thick layer for the upper layer. Note that for the increase in velocity, the variation is from +1 to +0.5 and for the decrease in velocity the variation is from −1 to −0.5.

velocity in the 300-m-thick layer. Taken together, a 300-m-thick layer with a decrease in velocity of −7% fits best the observed amplitude versus offset (Fig. 5).

**Frozen melt or partial melt sills.** The characteristics of the wide-angle reflection events, together with the synthetic test we conducted, suggest that the observed events could be either produced by the presence of frozen or partially molten sills in the mantle. A frozen gabbroic sill at 11–14-km depth would have a P-wave velocity of ~7 km s$^{-1}$, whereas the surrounding mantle peridotite would have a P-wave velocity of ~7.6–7.8 km s$^{-1}$, producing a velocity contrast of −8–10%, consistent with our observations. Frozen gabbroic sills have been suggested to occur in at mid-lithospheric depths (39–59 km) for older lithosphere (128–148 Ma) in the Pacific Ocean[18].

On the other hand, a melt channel has been proposed to occur at the base of 40–70-Ma old lithosphere in the Atlantic Ocean at 72–88-km depth[16]. The top of the melt channel, marked by a reflection with negative velocity contrast (−8.5%), is associated with 1260 °C isotherm, and is considered as a freezing boundary with solid lithosphere above and partially molten channel below. The bottom of the melt channel, defined by a deeper reflection with positive velocity contrast (+8.5%), corresponds to 1355 °C, and marks the top of the mantle asthenosphere. Although the top of the melt channel (base of the lithosphere) deepens with age following the 1260 °C isotherm, the thickness of the channel decreases with age, from 17 km at 40 Ma to 11 km at 70 Ma, indicating that the thickness of the melt channel should thicken towards the ridge axis[16]. One possibility is that the base of melt channel might correspond to base of the melting zone near the ridge axis, where the temperature must be higher[39]. Furthermore, based on these two positive/negative velocity contrasts, it was suggested that an average of 1.4% of melt might be present in the melt channel[16]. However, given the limitation of seismic reflection technique, it would be difficult to say with confidence if the top reflection corresponds to a thin melt sill or a thick melt channel. Therefore, the top reflection we image might be a partially molten sill with ~1% of melt at the base of the lithosphere and the deeper reflections might be partially molten sills within the melt channel as the base of melt channel would be much deeper (>70 km).

In conclusion, these reflections could either be partially molten or frozen sills, depending upon the temperature. As the solidus temperature of basaltic melt is ~1200 °C[40], we could use the 1200 °C isotherm to demarcate the boundary between the solid lithosphere above the melt channel.

## Discussion

Thermal regime near a ridge axis is influenced by several factors that include (a) asthenosphere upwelling beneath the ridge axis, (b) mantle flow stresses due to plate spreading, (c) melt segregation and associated mantle compaction, (d) horizontal extensional stresses in thickening lithosphere[2] and (e) hydrothermal circulation[3]. All these effects would tend to steepen the isotherms in the vicinity of the ridge axis and flatten them off axis, leading to a thick and nearly sub-horizontal base of the lithosphere near the ridge axis. As we move far away from the ridge, vertical thermal conduction will dominate, and the isotherm would follow the plate-cooling model and the lithosphere will thicken with age. The presence of nearly flat reflectors at 11 ± 0.5 and 14.5 ± 0.5 km below the seafloor spanning the lithosphere ages from 0.51 to 2.67 Ma seems to indicate that the ridge axis processes control the thermal structure up to at least 2.67 Ma.

On the intermediate-spreading JdF Ridge, an axial melt lens is observed at ~2–3 km below the seafloor[22], where the temperature

should be ~1200 °C, and hence the lithosphere just beneath the ridge axis would be 2–3-km thick. For purely a plate-cooling model (see the Methods section), the 1200 °C isotherm will cross the upper reflector at 1.25 Ma (Fig. 6a). On the other hand, if we take hydrothermal circulation into consideration[41] (see the Methods section) the 1200 °C will be deeper, and the upper reflector will lie between 800 and 900 °C isotherms, and the lower reflector between 900 and 1200 °C isotherms (Fig. 6b). So there are two possibilities:

1. The top reflection at 11 ± 0.5 km represents the base of the lithosphere, and the other deeper reflections could be melt sills within the melt channel (Fig. 6a), where shallowing and flattening of the lithosphere could be due to factors like mantle thermal anomaly, volatiles and sediment blanketing.
2. These reflections are produced by frozen melt sills in the lithospheric mantle, and the base of the lithosphere is deeper corresponding the 1200 °C isotherm, resulting in a thick young lithosphere (Fig. 6b).

The thinning of the lithosphere due to thermal mantle anomalies have been observed in the Pacific and Atlantic Oceans due to the presence Hawaiian Plume[42] and Cameroon Plume[43], respectively. In our study, the existence of several seamount chains on the west flank[20–22], the skewed N–S mantle low-velocity anomaly connecting the two OSCs[29] and the axis-centred 40-km-wide plateau associated with anomalous crustal thickening[22,23] indicate the presence of thermal anomaly near the ridge axis. These observations, and in particular the latter two suggest that this thermal anomaly may extend farther eastward from the ridge axis, and the flat reflection we observed up to 2.67 Ma old lithosphere might be due to this extended thermal anomaly. A subsidence analysis due to sediment loading using information from reflection seismic data[44] was done for the transect that crosses our survey area (Fig. 1a). It indicates an excess basement uplift of about 200 m extending up to 140 km (up to 3.7 Ma) from the ridge axis (Supplementary Fig. 14), which may indicate the presence of this thermal anomaly with a temperature ~30 °C higher than the surrounding mantle (assuming ~100-km-thick sub-plate asthenospheric channel and using the equation for estimating the regional uplift[45]). Therefore, a possible mantle thermal anomaly could be produced by a combination of hotspots on the Pacific Plate and north-westward migration of the JdF Ridge[46].

In order to explain the presence of melt channel below the lithosphere at 1260 °C in the Atlantic Ocean, the existence of water was invoked to reduce the solidus temperature[16]. If water is the main volatile in the melt channel at the base the lithosphere, for the upper reflections at ~11.0-km depth (~13.6 km below sea surface) at 1.5 and 2.5 Ma, a water content of 25 and 700 parts per million (ppm) would be required to reduce the solidus temperature, respectively (Supplementary Fig. 15)[16,47]. As the deeper reflection at 14.5-km depth is close to the 1200 °C isotherm, only at 2.5 Ma, some water (~100 ppm) would be required to maintain molten sills at these depths (Supplementary Fig. 15).

A significant part of the Endeavour segment and the young JdF Plate are covered with low-permeable terrigenous sediments[48], which can act as a thermal blanket, effectively limiting the hydrothermal circulation, and changing the thermal structure of the underlying lithosphere[49]. Drilling results from our study area[50] (Fig. 1b) suggest that the basement temperature is up to 60 °C higher with respect to a non-sedimented young oceanic lithosphere. The presence of sediments would have tendency to flatten the isotherm away from the ridge axis. Taken together, these factors could produce a thin lithosphere and a flat base of the lithosphere.

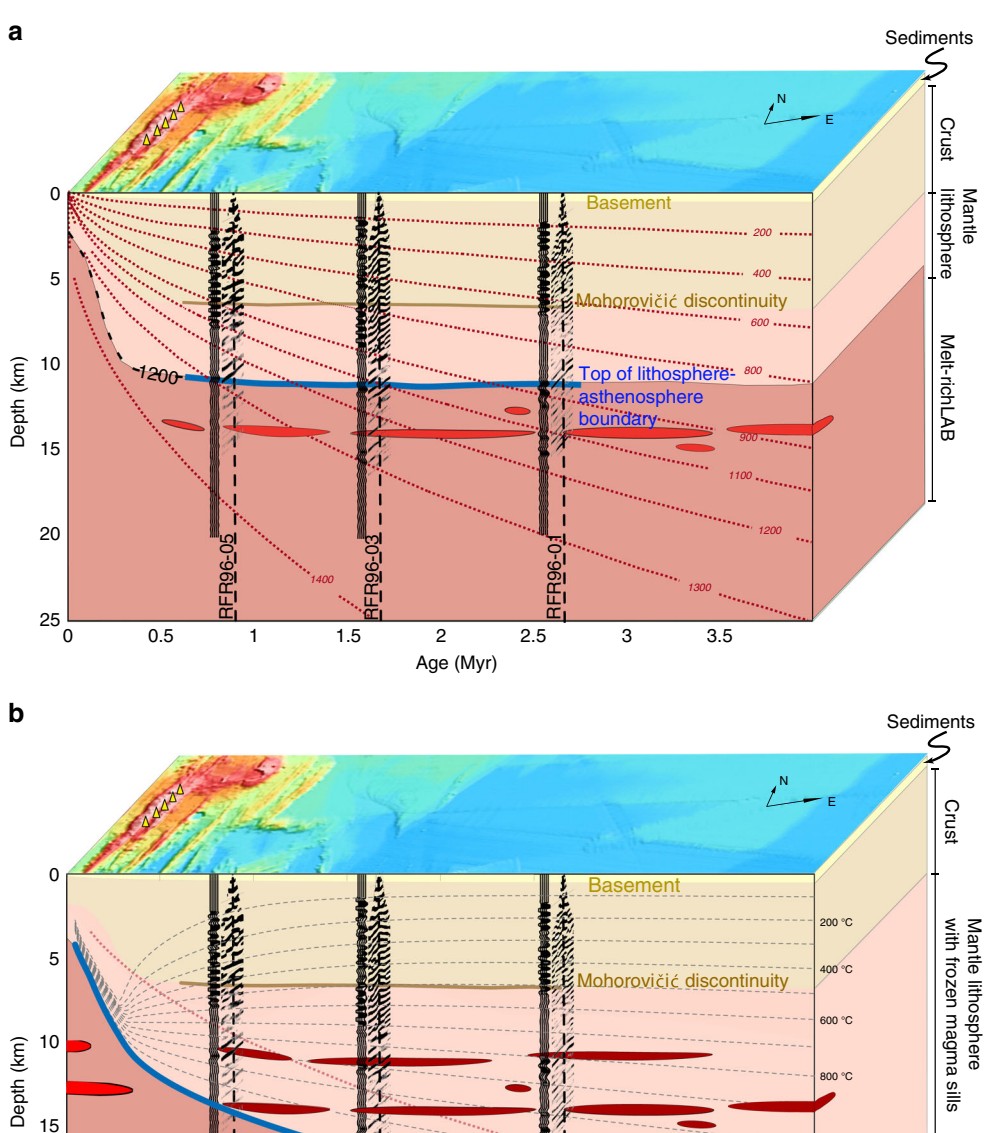

**Fig. 6 Thin lithosphere with flat lithosphere–asthenosphere boundary (LAB) versus thick lithosphere.** A schematic diagram showing two different models of the LAB. **a** Thin lithosphere and flat LAB: The top reflector imaged in the seismic sections (Fig. 3) is interpreted as the top of the LAB (blue). The deeper reflectors are interpreted as a presence of melt pockets (red ellipses). Pre-stack migrated traces at the crossing of three seismic profiles with the ridge perpendicular seismic reflection profile (black-white dashed line in Fig. 1) are shown. The red dotted lines represent isotherms calculated using the plate-cooling model[62]. Bathymetry is shown at an angle to add 'depth' to the two-dimensional illustration. The yellow triangles indicate hydrothermal vent fields. **b** Thick and heterogeneous lithosphere: The isotherms are represented by grey dashed lines obtained by the approach[3] described in Methods. The thick blue line marks the base of the thick and heterogeneous lithosphere. The red dashed curve is the 1200 °C isotherm based on the plate-cooling model in (**a**).

The S-wave receiver function results show the presence of a sub-horizontal converted S-wave receiver function image at ~30 km below the sea surface[14] underlying the JdF Plate between 0 and 8 Ma old lithosphere, suggesting that the flat base of the lithosphere might be a ubiquitous feature in this region. Furthermore, a S-wave tomography study suggests that the S-wave velocity is higher than expected for a plate-cooling model, indicating the presence of a colder and thicker lithosphere[9]. Assuming an S-wave velocity

contour of 4.4 km s$^{-1}$ for the base of the lithosphere at 48°N on the JdF Plate, tomographic results indicated that the lithosphere should be ~28-km thick near the ridge axis, increasing rapidly to ~38 km thickness farther from the ridge axis and then remain constant up to the subduction front[51], further supporting the idea of a flat but thick lithosphere. An excessively thick young ocean lithosphere has also been detected in a magnetotelluric study east of the Mohns Ridge in the northern Atlantic Ocean[52].

The presence of the extended frozen melt sills over a 25 km by 65 km area in a 0.5–2.65 Ma old lithosphere requires the formation of these sills to be very close to the ridge axis. As the thermal gradient near the ridge axis is very high (Fig. 6b), melt sills present at $11 \pm 0.5$ km, and $14.5 \pm 0.5$ km depth beneath the ridge axis could freeze across the 1200 °C isotherm very rapidly, forming extended frozen melt sills in the upper lithosphere. Recent studies have suggested that the LAB is controlled by an impermeable boundary related to melt crystallization preventing melt migrating upward, representing a freezing boundary[16,52,53]. Therefore, in order to have frozen melt sills extending over 25 km along the axis and more than 65 km away from the ridge axis, the molten melt sills must be in a steady state at $11 \pm 0.5$ and $14.5 \pm 0.5$ km depths along a significant part of the spreading centre.

The presence of frozen basaltic melt sills would make the mantle lithosphere chemically heterogeneous. The chemical heterogeneity of the mid-ocean ridge basalt (MORB) has been known for a long time[54]. It has been suggested that about 10% of recycled MORB are present in the bulk mantle. Furthermore, the mantle source comprises a small amount of (10%) eclogites or pyroxenite hosted in ambient peridotite mantle[54] that would be more fusible than the host peridotite. The heterogeneous mantle rich in fusible components will melt easily and will create a magmatic channel that could extend up to the base of the lithosphere and freeze as the lithosphere cools[54], forming the frozen melt sills we observe.

Surface wave tomography results indicate the S-wave velocity is higher than normal east of the JdF Ridge, indicating the absence of a thermal anomaly in our study area[9]. Furthermore, a recent high-resolution surface wave study[51] indicates a 28-km-thick lithosphere near the ridge axis. Taken together, all these results support the idea that the reflections we have observed are frozen melt sills, and the lithosphere is heterogeneous and thick. If oceanic lithosphere is heterogeneous, one should be able to image frozen melt sills elsewhere in ocean basins using existing wide-angle data.

However, the presence of a thermal anomaly in the mantle and a thin lithosphere could not be ruled out. Surface wave tomography results west of the JdF Ridge show a low S-wave velocity anomaly, indicating the presence of thermal anomaly in the mantle, supported by the existence of the chain of seamounts. The thickened crust near the ridge axis and the high attenuation near the ridge axis[55], support the idea of the mantle thermal anomaly.

The thermal evolution of thin and thick lithosphere with age would be very different, and especially their roles in the subduction process at the Cascadia subduction zone. A thin and warm slab would be buoyant, and hence would lead to a shallow dipping slab, confirmed by the low dip (10–12°) of the subducting plate observed along the Washington segment of the Cascadia subduction[56]. In contrast, a thick heterogeneous lithosphere would be cold and hence would subduct easily.

## Methods

**Seismic data acquisition**. Twenty-four ocean-bottom hydrophones (OBH) were deployed along four ridge-parallel profiles. Along each profile, six OBHs were deployed at a 4-km interval over 20-km distance (Fig. 1). OBH525 did not provide any data along profile RER96-05, and OBH534 was offline of profile RER96-08, and hence was not used in the data analyses. The shots were fired along ~50-km long profiles, providing maximum offsets of ~35 km (Fig. 1b).

Except for profile RFR96-05, the data were acquired using a single BOLT Inc. airgun of 32 l, fired at 140 bar, towed at 15-m water depth. The dominant frequency of the source was 6 Hz. For profile RFR96-05, a PS100 Sleeve gun (60 l) was used instead. The shot spacing was 40 s (equivalent to ~90 m). OBH 501–506 were deployed along profile RFR96-01, OBH 511–516 along profile 96-03, OBH 521–526 along profile 96-05 and OBH 531–536 along profile 96-08.

The data were band-pass filtered between 2 and 15 Hz. A predictive deconvolution was performed to enhance the deep crustal and mantle arrivals. Strong crustal arrivals (Pg) are observed to an offset of 22.5 km, which are followed by wide-angle reflections from the Moho (PmP). As the maximum offset is 35 km, no mantle arrivals (Pn) are observed in the data. Representative examples of OBH data are shown in Fig. 2 and Supplementary Fig. 1. Apart from the crustal arrivals, one can also observe wide-angle reflection arrivals, originating within the mantle on most of the OBH gathers, on both sides of the OBH. This reflection generally cuts across P-to-S converted energy.

**Tomography**. Travel times of first arrival P-waves have been hand-picked with picking uncertainties of 20–30 ms for short-offset P-waves (Pg) and 40–60 ms for secondary wide-angle reflected arrivals (PmP and sub-Moho mantle reflection). We applied a joint first arrival refraction and reflection tomography using a hybrid ray-tracing scheme combining the graph method with further refinements utilizing ray bending with a local conjugate gradients method for inversion[57]. Smoothing and damping constraints regularize the iterative inversion.

For the starting model, we used a 1D velocity for the oceanic crust characterizing the area[58] and hung it below the basement defined by coincident seismic reflection data[59,60]. The inversion was carried out using a top-down approach, inverting first for the shallow structure and adding additional arrivals while inverting for crustal thickness and velocity structure. Later, we inverted for the depth to the upper mantle reflector. In general, the inversion results provided smooth velocity models with a root-mean square misfit of 35–50 ms.

We carried out a nonlinear Monte Carlo type error analysis to derive model uncertainties[61]. The mean of the model gives the average velocity and the variance deviation from the average. The best-fitting model is shown in Supplementary Figs 2, with the uncertainty and derivative weight sum. The final model had root-mean-square misfits of 40–60 ms and Chi-square <1.

In addition, we performed tomographic inversion to fit the deep-seated wide-angle reflections, as shown in Supplementary Fig. 3. Here, we assumed that the upper mantle velocity increases with age as has been observed along the East Pacific Rise[62] and included the mantle discontinuity as a floating reflector, inverting the geometry and depth of the reflector.

**Pre-stack depth migration**. The pre-processing of the data includes a predictive deconvolution and band-pass filtering (2–15 Hz), followed by a spherical divergence compensation. The OBH geometry places the shots and receivers on a different datum. To relocate the shots and have them at approximately the same datum level as the receivers, we applied a wave equation datuming[33,34,63,64] to a constant depth, which is the shallowest point of the seafloor depth along a profile. Finally, to compensate for the small elevation differences between sources and receivers, we applied static corrections.

The spatial aliasing caused by the downward continuation was removed by dip filtering (5–6°). Similarly, the Pg and P–S-converted waves were removed as they could affect the migration of mantle reflection signals. The water bottom multiples were also muted. The resulting receiver gathers were migrated using the Kirchhoff pre-stack depth migration technique[31,32]. Travel times for the migration were calculated by ray tracing through the velocity model obtained by the tomography for profile RFR96-01. The velocity model was sampled on a 30 m × 30 m grid for the migration. The final migrated gathers were summed, and the seismic images are displayed in Fig. 3 and Supplementary Fig. 5. To make sure that the migrated images are real, we also stacked the normal moveout (NMO) corrected downward continued data, and post-stack time migrated them (Supplementary Fig. 7). The NMO velocity was estimated from the tomographic velocity model. The correspondence between the post-stack migrated image and the NMO corrected OBH gathers (Supplementary Fig. 7) confirms that they are real images from the mantle.

**Synthetic seismogram modelling**. In order to make sure that the deep reflection arrivals are real, not multiples, and to quantify the nature of the reflection interface, we computed synthetic seismograms (Supplementary Fig. 4). Based on the travel time tomography and velocity model obtained for profile RFR96-01 (Supplementary Fig. 3d), we extrapolated the velocity model down to 20-km depth. To simulate the mantle reflectors, we inserted low- and high-velocity layers in the mantle of different thicknesses and velocity contrasts.

We computed the synthetic seismograms by solving the 2-D isotropic elastic wave equation by fourth-order finite difference in space and second-order in time using the staggered grid method[65]. The first model is a 4-km-thick low-velocity zone where the velocity is decreased by −7%. The modelling results confirm that these reflections are real, not multiples (Supplementary Fig. 4).

The next step was to determine if the reflection is due to high or low thin velocity layers. We included 300- and 600-m-thick low- and high-velocity layers with a velocity contrast of ±7% for the upper layer (Supplementary Figs. 10–12).

Extensive synthetic seismogram modelling was carried out by varying the thickness of the upper layer from 300 to 1200 m (Supplementary Fig. 10), the velocity variations from ±7 to −30% (Supplementary Figs. 11 and 12), and also allowing a gradient in the velocity (Supplementary Fig. 13).

These results suggest that the first mantle reflection is related to a negative velocity contrast, or a negative velocity gradient within a thin zone (<1/4 of wavelength). Similarly, the deeper reflections require the presence of thin low-velocity layers.

**Amplitude versus offset**. We computed the amplitude versus offset variation for real as well synthetically computed data for different velocities in a 300-m-thick layer (Fig. 4). A curve was fitted to obtain a smooth amplitude versus offset curve. In order to compare different amplitude versus offset variations for different cases, the amplitude versus offset curve was normalised by the amplitudes at 8.5-km offset for each case, creating a relative amplitude versus offset plot (Fig. 5).

**Computation of isotherms**. The isotherm in Fig. 6a were computed using the plate-cooling model[66,67] for a spreading rate of 60 mm yr$^{-1}$, and a mantle temperature of 1350 °C at the base of the lithosphere at 106 km and thermal diffusivity $\kappa = 1$ mm$^2$ s$^{-1}$.

The isotherms in Fig. 6b were computed using the approach that incorporates extensive hydrothermal circulation near the ridge axis and the plate-cooling model away from the ridge axis[4]. A key element of this simplified model is that the crust cooled near the axis has to be heated from below before near-surface temperature gradients. As a result, the isotherms will deepen rapidly near the spreading axis and flatten away from the ridge axis. This model treats only the vertical conduction of heat through the lithosphere as it moves away from a spreading centre. The effect of hydrothermal convection near the spreading axis is approximated by the enhancement of thermal diffusivity near the axis. The enhancement factor, termed the Nusselt number, gives the ratio of the steady-state average vertical heat transport through a convecting layer to the heat transported by conduction alone.

An explicit finite-difference approximation of the one-dimensional, time-dependent equation for vertical, conductive heat transport is used to solve lithospheric temperatures as functions of time $t$ since the formation of the lithosphere and depth $z$ below the seafloor. We assume the initial temperature of the crust and mantle at the ridge axis, to be 1200 and 1350 °C, respectively. Latent heat is liberated as the crust cools between its liquidus temperature $T_L = 1150$ °C and its solidus temperature $T_S = 1100$ °C. The total latent heat $L$ released over this temperature range is set to 400 kJ kg$^{-1}$ (ref. [66]) and the assumed value of the specific heat $c$ of 1.1 kJ kg$^{-1}$ °K$^{-1}$, consistent with measurements for basalt[68]. The resulting value of $L/c = 360$ °K, which is the change in sensible temperature that would produce the same release of heat as through the latent heat release.

The thermal diffusivity is set to an effective value of $Nu$ times $\kappa$, where $\kappa = 5 \times 10^{-7}$ m$^2$ s$^{-1}$ appropriate for basalt[69] or olivine at temperatures above ∼500 °C[70]. From time $t = 0$ to $t = t_{OFF}$ and from $T = 0$ to $T = T_{OFF}$ the $Nu$ is set to the on-axis value of $Nu_{ON}$. The idea behind setting a maximum temperature for hydrothermal flow effects is that ductile flow should close cracks above a rheologically controlled temperature. For $t > t_{OFF}$ or $T > T_{OFF}$ then $Nu = 1$. We obtain temperature changes by solving a finite-difference version of:

$$\frac{\partial T(z,t)}{\partial t} = \frac{\partial}{\partial z}\left(Nu\,\kappa\,\frac{\partial T(z,t)}{\partial z}\right) - H_L(T) \quad (1)$$

where

$$H_L(T) = 0 \text{ for } T > T_L \quad (2)$$

$$H_L(T) = \frac{L}{c(T_L - T_S)} \text{ for } T_S < T < T_L \quad (3)$$

$$H_L(T) = 0 \text{ for } T < T_S \quad (4)$$

relates to the release of latent heat. Temperatures are laterally advected at the plate velocity $V_P$ so that the horizontal position $x$ can be related to time as $x = V_P t$. The top boundary was kept at 0 °C and the bottom, at 60-km depth was kept at 1350 °C. A grid size of 200 m was used.

We varied $Nu_{ON}$, $t_{OFF}$ and $T_{OFF}$ to see if we could produce a fairly flat isotherm at ∼11 km out to ∼3 Ma as indicated by the seismic data for the flank of the Juan de Fuca Ridge. The case shown in Fig. 6b assumes the rather extreme values of $Nu_{ON} = 100$, $t_{OFF} = 0.3$ Ma and $T_{OFF} = 100$ °C but decreasing $Nu_{ON}$ and increasing $T_{OFF} = 100$ °C gives similar lithospheric thickening with time.

## Data availability

The following figures are based on the raw data used in this study: (1) Figures 2, 3 and 5 in the main text. (2) Supplementary Figs. 2, 3, 4, 5 and 7. The raw ocean-bottom seismometer data are available on the German PANGAEA database. The processed data shown in Fig. 3, Supplementary Figs. 5, 7 and 8 the German PANGAEA database.

## Code availability

We have used two types of codes: (1) to model synthetic seismograms shown in Fig. 4, Supplementary Figs. 4, 9, 10, 11, 12 and (2) to compute isotherms in Fig. 6b. These codes are propriety code but will be available on request from Singh (synthetic seismogram modelling code) and Buck (isotherm computation code).

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

## Acknowledgements

We thank Heinrich Villinger and Fares Mehouachi for helpful discussions and valuable suggestions that substantially improved the paper. We are grateful to Heinrich Villinger, chief scientist of the R/V Sonne cruise SO111 in 1996. We would like to thank Tim Stern and two anonymous reviewers for their constructive reviews, which improved the paper significantly. Funding from the Germany Science Foundation (DFG grant GR1964/2-1) and German Federal Government (BMBF grant 03G0111A) is acknowledged. The research leading to these results has received funding from the European Research Council under the European Union's Seventh Framework Programme (FP7/2007-2013)/ ERC Advance Grant agreement n° 339442_TransAtlanticILAB. The work is IPGP contribution number 4160.

## Author contributions

Y.Q. processed and analysed the seismic data and participated in the writing of the paper. S.C.S. supervised the project and wrote the paper. I.G. recognized the deep-seated mantle reflections in deconvolved record sections performed seismic tomographic inversions and contributed in the writing of the paper. M.M. participated in the writing of the paper. R.B. performed the thermal modelling.

## Competing interests

The authors declare no competing interests.
