## [Peer Review File · Nature Communications]

Reviewers' comments:

Reviewer #1 (Remarks to the Author):

Review of Thin Lithosphere and Mantle Thermal Anomaly beneath the Young Juan de Fuca Plate

By Qin et al

This paper presents a new seismic analysis of data from the Juan de Fuca ridge to show evidence for sub-horizontal reflectors defining the base of the lithosphere (LAB) and that these reflectors are best described as melt bodies. They see two distinct reflections separated by a distance of about 5 km. They note previous interpretations of a double reflection at LAB depths to indicate a melt channel. They tend to favour that interpretation for this study, but note that the depth to the LAB does not vary even though the age of the plate varies from 0.5 to 2.67 Ma. This observation they attribute to a smearing of the thermal anomaly due to hydrothermal circulation near the ridge axis. They also note the LAB is shallower than predicted and that to get melt in a channel at these depths extra water would be required to buffer the melting temperature down.

Overall, I think the authors make a good case for their observations and interpretation. I am familiar with looking at deep mantle reflections and the ones they show look convincing. i.e. in figure 2 and fig. S1. The migrated stacked sections were not quite so clear but I understand why. Their explanations for the depth range of the proposed LAB reflections and the flatness of the LAB also seem plausible.

The last paragraph of the paper (lines 234-242), on thickness of lithosphere and size of megathrust earthquakes, seems a little bit out of left field, but I understand the point the authors were trying to make. But the point is not well made in that last paragraph. Either re-word or drop it out and just focus on the LAB finding.

Overall I found this paper interesting, well written and full of new observations. It is this sort of result that will influence our thinking about plate structure and plate tectonics in general. My recommendation is that your journal should publish this with minor modifications.

Some editorial detail:

Line 20 : after word “extending” it needs “laterally out to an effective age of”

Line 22: insert “a” after “Such”.

Line 118: the use of the term “downward continued”. This is a term used with potential fields, but is it correct for what you do in seismology? My understanding is that its just a delay time correction ? or is it really a downward continuation ?

Line 147: I could not see in fig. 2 the reverse polarity ? maybe this detail should be in supplementary material with some more explanation ?

Line 160 : The attribution of the melt layer at the LAB, based on seismic reflections, should be to reference # 11. They had the first paper on this , and reference 12 followed.

Line 165: there is a word or two missing here ?

Line179: The attribution of an idea to a third person as a pers comm does not work here. You should cite a paper that backs this idea ? or just say “ We argue that....”

Lines 234-242: as discussed above this is the major bit of the paper that needs re-crafting or being dropped.

Lines 260 267: I wonder if F-K filtering could have been used to enhance the mantle reflections ?

Lines 435: there is a mix up in the labelling of S2 and S3. ?

Tim Stern

Victoria University of Wellington

Reviewer #2 (Remarks to the Author):

The authors image reflectors at 11 and 14 km depth beneath 0.51 to 2.67 My old seafloor east of the Juan de Fuca Ridge over a 65 X 25 km lateral area. These are interpreted as related to an isolated melt pocket at the base of the lithosphere, within a larger melt channel. The fact that it does not increase in depth with age is interpreted as related to anomalous temperatures.

This kind of high resolution observation is important to our understanding of the oceanic lithosphere, and therefore plate tectonics in general. Observations constraining oceanic lithosphere at sub-Moho depths are extremely rare. Therefore, the work could be novel and of interest.

However, for a full understanding of its importance and significance, the authors need to do a better job quantifying what is required by the data, and clarifying their writing and arguments. The finding should then be better placed in the context of previous work globally and in this region. As written the interpretation and conclusions are confusing and not well-supported.

Major comments:

There is no quantitative estimate for the required velocity contrast in the main text. It simply states (line 149): The velocity contrast must be large. The result itself should be more quantitatively presented, and quantitatively put in the context of previous work, with an interpretation that directly takes into account the possible scenarios, errors, etc.

Line 159 The authors state that the observation could be a melt-rich channel or an isolated melt pocket in a larger channel. The reason states for choosing the latter is that Mehouchi & Singh hypothesized channels should get thicker by the ridge. This interpretation was far too abrupt and conjectural. Much more logic/explanation is needed in the main text, first, why it needs to be the LAB, and second, for the melt pocket idea. These reasons should be related to the current results, not previous hypotheses. In the Supplemental models with 2 velocity notches are shown, for 2 melt lenses, but this is not the way the story is presented in the main text, so not very connected.

What about anisotropy? Could this influence and/or explain your result? Many papers have attributed apparent lithospheric discontinuities to anisotropy (including for example, Audet et al., 2016 – at the Juan de Fuca Ridge) .

The presentation of the interpretation is confusing. Only after looking at the supplemental material was it clear that the 2 features (at 11 and 14 km depth) are being interpreted as melt pockets. In the main text the wording is never presented that way. There are a range of thicknesses depending on where you are reading, which was confusing:

Line 137 – the thickness is 4 km thick

Abstract and many other place : $14 - 11 = 3$ KM Thickness?

Line 310 – 311: modelling of layers 0.3 – 0.6 km thick?

The abstract reads:

“We interpret the upper reflection as the base of the lithosphere and the lower reflection as thin melt sheet within the lithosphere-asthenosphere boundary channel.”

It was not clear that the second phase came from another melt sheet. If you change to: “We interpret the 2 reflections as caused by 2 thin (X km thick) melt sheets within a melt-rich lithosphere-asthenosphere boundary channel” That would be much clearer.

If a range of scenarios is possible (melt lenses, thick layers, etc.), that is fine, but it should be clarified in the main text what is/is not acceptable, rather than presenting different numbers/scenarios in different sections.

Also, are you seeing phases from the top and the bottom of the sheets?

Why can you image individual melt sheets here in Cascadia, but only imaged the top and the bottom of the LAB channel at the mid-Atlantic Ridge? This should be made clear in the text.

The modelling in Fig. S4 should be more quantitative in terms of comparison to data, and should be discussed much more in the main text. As is the result presented in the main text is not quantitative in terms of what structure is required by the data.

Line 152 – 152 – No petrological mechanism exists to give a high velocity layer? What about frozen-in ultra mafics? I also found it curious that there was no reference to [Ohira et al., 2017 Evidence for frozen melts in the mid-lithosphere detected from active-source seismic data (Scientific Reports)]. In this paper a very similar observation was interpreted as a frozen-in MLD.

It is good that the authors compared to the receiver functions, but there has been a lot of other excellent work in this particular region of Cascadia (by Bell & Forsyth; Toomey; Gao; Audet; and many others), but references and comparisons to these – where appropriate - were lacking. The result could have been put into context better.

Line 170: The authors state that the lithosphere thickens along the 1200 degree isotherm in the plate cooling model. This is not true/proscribed by that model.

The reasons for the flatness of the interface are presented in a confusing way, without enough logic, support detail.

Line 173 – 176: To explain the flatness of the reflector, the authors call on hydrothermal circulation, then state that it would not be enough, or only help to 0.3 My with a reference to Roger Buck personal communication (?!). This is far too casual. More logic, explanation, background is required. Include calculations with references would be ideal. Also, why is hydrothermal needed? At intermediate spreading centers a thickened plate is predicted beneath the ridge. More logic is needed.

Then the authors call on a sediment blanket to increase heat at older ages caused by increased terrestrial sediments. However, most of the terrestrial sediment gets stuck on the other side of the trench, so not clear this is true. Estimates for sediment thickness near the ridge are typically < 1 km. More logic/support/detail would help.

Line 209: the mechanism for flattening described in this section is hotspots, but without a clear link to the previous interpretation (sediment). More information about what is actually required by the data would help, or even more connection between the interpretations.

Line 211 – 212 not sure how hotspot anomalies get sheared or dragged?

Line 230 onwards has a lot of strong statements about implications, but most are too brief/terse. All require a few more sentences of logic to connect.

Minor comments:

Line 33, 40, and 2017: these statements are vague and overly negative to surface waves and receiver functions. Surface waves lateral resolution is not always 100s km – it is smaller in regional experiments and at shallow depths, so that statement is incorrect. There is a reference to an Eaton review paper, but not clear what Eaton was talking about, maybe global studies of much deeper depths (?). It would probably better to go with a surface wave paper that does testing and make the statement more specific. Also, receiver function depth resolution is typically better than 10 – 15 km at lithospheric depths, so again incorrect, and the statement would need a reference to back it up.

Line 38 – 39 Are the receiver function depths below sea surface or seafloor? This is Important for comparison.

Lines 213 – 215 need to be referenced.

Line 208- 209 this is a vague comparison. It should be more specific and quantitative.

Reviewer #3 (Remarks to the Author):

Comments on the manuscript “Thin Lithosphere and Mantle Thermal Anomaly beneath the Young Juan de Fuca Plate” by Yanfang Qin et al.

The authors develop a novel method to generate seismic reflection images from wide-angle seismic data. This method is applied to seismic data collected in the Endeavour segment of the Juan de Fuca Ridge. The results show reflections in the mantle, which are interpreted as the base of the lithosphere at 11 km depth and melt sill lenses at 14 km. The reflections remain flat as a function of plate age, between 0.5-2.7 Myr, which leads the authors to invoke a mantle thermal anomaly.

I was left confused by the interpretation presented in this manuscript. Maybe it is a problem of organization, but there appear to be conflicting statements and assumptions being made that left me questioning the validity of the conclusions. The authors should address or clarify the following:

Using wide-angle data to generate a pre-stack depth migrated (PSDM) reflection image is an interesting idea. But this method deserves a more thorough analysis than is presented here, specifically to demonstrate the strengths and pitfalls of the approach. The synthetics presented in Fig S4 are a promising start. At the least, the authors should provide some examples of what happens when the synthetic seismograms shown in Figure S4 are processed using their PSDM approach. How does an imperfect assumed velocity model distort the reflection image? Etc.

L109. An age and hence temperature dependent mantle velocity is applied in the PSDM, where velocity increases with plate age. Does this not conflict with later argument that the mantle

isotherms are flat? The relative velocity differences are small and may not change the imaging results much, but this should be corrected or justified.

L185. This paragraph appears to contradict the preceding paragraph and makes an unphysical claim. The authors first invoke vigorous hydrothermal circulation in order to rapidly depress the mantle isotherms and flatten them away from the axis for a short distance (~0.3 Myr equivalent). Then, the authors refer to drilling results that support much warmer basement temperatures relative to unsedimented young seafloor. If significant heat is already removed near the ridge axis then adding sediment and stopping the advective heat loss may warm the basement temperature, but it does not increase mantle temperatures. It simply slows down the rate of mantle temperature decay with age. I think the authors intended to say that in addition to sedimentation cutting off hydrothermal circulation, an unrelated heat source is required that flattens the 1200 degC isotherm by offsetting conductive heat loss as the plate ages. However, ref. 37 directly states that the heat flow on the sedimented JdF Plate in the Endeavor segment matches predictions from plate cooling models. Flattening the 1200 degC isotherm to the point where temperatures are warmer than plate cooling predictions should yield anomalously high heat flow, which is not the case here. Not only that, the heat flow would be constant with age to reflect a flat isotherm (e.g. no change in temperature with depth), again this is not the case. Sediment cover merely limits cooling to conductive heat loss.

L199. The authors state that the subsidence analysis suggests 30 degC warmer mantle. Is this enough to sufficiently shallow the isotherms between 0.5 and 2.7 Myr? Intuitively seems unlikely, and does not appear to be the case based on isotherm contours in Fig 3 as well. How would warmer background mantle manifest as flat isotherms? Furthermore, the subsidence analysis does not consider the effect of crustal thickness variations on basement depth, which is critical when inferring warmer mantle (based on isostasy, thicker crust equals shallower seafloor and hence no thermal anomaly is needed). The gravity anomalies in the region cannot be ignored either.

L224. Can the authors distinguish between melt pockets and frozen sills? Is it not simpler to interpret the reflectors as frozen sills and thereby avoid the complex and seemingly inconsistent thermal interpretation required by melt?

L229. Again, sediment blanketing does not increase mantle temperature, heat is still lost via conduction. Indeed, plate cooling models ignore hydrothermal circulation and only consider conductive heat loss that is representative of sedimented oceanic plates. This leaves a 30 degC mantle temperature increase inferred from the subsidence analysis in Fig S8, which as previously noted is problematic in itself and requires additional justification.

Minor points:

L122. This statement is confusing. Why does the downward continued data, which is muted, yield reverberations?

L136. What exactly are the uncertainties in mantle velocity structure? This relates to a previous point regarding L109.

L155. What is the minimum V_p reduction required to generate the reflections? Is it 8.5%?

L200. Authors should explicitly say that they are using the subsidence analysis described in ref. 38 to avoid confusion.

Reviewer #1 (Remarks to the Author):

Overall, I think the authors make a good case for their observations and interpretation. I am familiar with looking at deep mantle reflections and the ones they show look convincing. i.e. in figure 2 and fig. S1. The migrated stacked sections were not quite so clear but I understand why. Their explanations for the depth range of the proposed LAB reflections and the flatness of the LAB also seem plausible.

The last paragraph of the paper (lines 234-242), on thickness of lithosphere and size of megathrust earthquakes, seems a little bit out of left field, but I understand the point the authors were trying to make. But the point is not well made in that last paragraph. Either re-word or drop it out and just focus on the LAB finding.

We agree with reviewer and we removed the discussion on the link between the LAB and earthquakes, but have left a couple of sentences on the role of the thick versus thin lithosphere in the subduction process.

Overall I found this paper interesting, well written and full of new observations. It is this sort of result that will influence our thinking about plate structure and plate tectonics in general. My recommendation is that your journal should publish this with minor modifications. Some editorial detail:

Line 20: after word “extending” it needs “laterally out to an effective age of”

We have rewritten this sentence and have been made clear.

Line 22: insert “a” after “Such”.

Corrected.

Line 118: the use of the term “downward continued”. This is a term used with potential fields, but is it correct for what you do in seismology? My understanding is that its just a delay time correction ? or is it really a downward continuation ?

We have defined the word ‘downward continuation’ and have linked with ‘re-datum’, which is a standard technique in seismic terminology. We also brought the text from the Methods section to the main text to explain more details of what was done prior to migration.

Line 147: I could not see in fig. 2 the reverse polarity ? maybe this detail should be in supplementary material with some more explanation ?

We agree with the reviewer and now show polarity of raw data in Figure 2. As indicated in the text, we find both polarities and hence the results are non-conclusive, which we discuss in the text.

Line 160 : The attribution of the melt layer at the LAB, based on seismic reflections, should be to reference # 11. They had the first paper on this , and reference 12 followed.

We have taken this point into account in the revised version.

Line 165: there is a word or two missing here ?

We have rewritten this section.

Line179: The attribution of an idea to a third person as a pers comm does not work here. You should cite a paper that backs this idea ? or just say “ We argue that....”

We have included Roger Buck as a co-author and the corresponding figure has been included in Figure 6B.

Lines 234-242: as discussed above this is the major bit of the paper that needs re-crafting or being dropped.

As mentioned above, we have removed the discussion on earthquakes.

Lines 260 267: I wonder if F-K filtering could have been used to enhance the mantle reflections ?

We tested FK filter and found the results were not very promising (see attached figure), and therefore, we did not use them. Since we want to pre-stack migrate the data, any noise introduced during FK filter would create artifacts in migration. Therefore, we used the least noisy data for imaging purpose.

Figure 1 – A) Observed data recorded on OBH 501. B) f-k spectrum of the original data. C) Data obtained using F-k filter (note the cross-like artifact at ~2 s). D) F-k spectrum of filtered data.

Lines 435: there is a mix up in the labelling of S2 and S3. ?
We have addressed this point.

Reviewer #2 (Remarks to the Author):

Major comments:

There is no quantitative estimate for the required velocity contrast in the main text. It simply states (line 149): The velocity contrast must be large. The result itself should be more quantitatively presented, and quantitatively put in the context of previous work, with an interpretation that directly takes into account the possible scenarios, errors, etc.

We appreciate Reviewer's concern and have tried to address these points first by discussing more the modelling results in the main text and then using the amplitude versus offset analyses to quantify the velocity contrasts at the interface. We have also added new figures in the main text (Figures 4 and 5) to address these points. However, we would like to mention that the low signal to noise ratio of data hinders really saying with confidence the velocity contrast required by the data. If we have a continuous long offset seismic reflection data, which are more prone to amplitude versus offset analysis or full waveform inversion, we might be able to quantify the nature the reflections better.

Line 159 The authors state that the observation could be a melt-rich channel or an isolated melt pocket in a larger channel. The reason states for choosing the latter is that Mehouchi & Singh hypothesized channels should get thicker by the ridge. This interpretation was far too abrupt and conjectural. Much more logic/explanation is needed in the main text, first, why it needs to be the LAB, and second, for the melt pocket idea. These reasons should be related to the current results, not previous hypotheses. In the Supplemental models with 2 velocity notches are shown, for 2 melt lenses, but this is not the way the story is presented in the main text, so not very connected.

We now give different possible models, and explain the reason behind our preferred model. We also discuss the two end member models in detail. Furthermore, we have brought modelling in the main part of the text be consistent with the modeling and interpretation.

What about anisotropy? Could this influence and/or explain your result? Many papers have attributed apparent lithospheric discontinuities to anisotropy (including for example, Audet et al., 2016 – at the Juan de Fuca Ridge) .

Our results show the presence of wide-angle reflections, which require sharp velocity contrasts (vertically). Though anisotropy might be present on larges-scale, on short wavelength scale one would require lithological variations, e.g. frozen melt, dunite, or molten melt etc.

The presentation of the interpretation is confusing. Only after looking at the supplemental material was it clear that the 2 features (at 11 and 14 km depth) are being interpreted as melt pockets. In the main text the wording is never presented that way. There are a range of thicknesses depending on where you are reading, which was confusing:

We agree with the reviewer and have tried to clarify this point. The confusion arising from the fact that pre-stack image is wrt to the seafloor whereas models are wrt to sea surface. We have used only one set of depth.

Line 137 – the thickness is 4 km thick

Abstract and many other place : $14 - 11 = 3$ KM Thickness?

Line 310 – 311: modelling of layers 0.3 – 0.6 km thick?

The first two cases represent the depths at which the two prominent reflectors are observed. They represent average values and to be consistent throughout the text we modify “~4 km” in line 137 to “~3 km”. However, the last example provided by Reviewer represents the thickness of the layers we used in our modes from which our reflection signal could potentially originate. We modified the caption of Figure S4 accordingly.

The abstract reads:

“We interpret the upper reflection as the base of the lithosphere and the lower reflection as thin melt sheet within the lithosphere-asthenosphere boundary channel.”

It was not clear that the second phase came from another melt sheet. If you change to: “We interpret the 2 reflections as caused by 2 thin (X km thick) melt sheets within a melt-rich lithosphere-asthenosphere boundary channel” That would be much clearer.

We have rewritten the abstract and have modified the text as suggested by Reviewer.

If a range of scenarios is possible (melt lenses, thick layers, etc.), that is fine, but it should be clarified in the main text what is/is not acceptable, rather than presenting different numbers/scenarios in different sections.

As mentioned above, we have brought the modelling section in the main paper, and have discussed different scenarios, accepting and rejecting the models, then performing amplitude versus offset analyses, before interpreting them.

Also, are you seeing phases from the top and the bottom of the sheets?

Our synthetic tests show that we should be able to observe the top and bottom reflection of a low velocity layer (with 7% assumed velocity reduction) when the layer is more than 500 m thick. For instance, in the panel A and D (Figure S11), the assumed thicknesses of the reflector shallower reflector are 600 m and 1200 m, respectively. Both of the reflections display a complex reflection signal in the modelled gather that can be identified as the top and the bottom with clear polarity reversal. However, for the cases of velocity layers lower than 500 m, as it shown in panel G (Figure S11) the reflection from top and bottom are not observed, i.e., there is superposition of the signal from the two arrivals.

Why can you image individual melt sheets here in Cascadia, but only imaged the top and the bottom of the LAB channel at the mid-Atlantic Ridge? This should be made clear in the text.

First, the absence/presence of melt sills could be related to the different data we are using. While, the study of Mehouchi and Singh (2017) uses active source multi-channel reflection seismic data collected using 12 km long streamer, here we use a wide-angle OBH data. The imaging power in both techniques also decreases with depth (11 km below the seafloor here as compared to 70-80 km in the Atlantic Ocean). Thus, the melt lenses (or pockets) could be present within the LAB channel. In fact, if we look carefully in the MCS (Mehouchi and Singh, 2017) data one could potentially identify additional weaker events. However, given the S/N it is difficult to argue if they are real or artifacts, i.e., the lenses may exist but we may not see them due to limitations of MCS data.

Second the absence/presence of the melt lenses could be attributed to important differences arising from geological settings of these two areas, the MAR is a slow spreading and Juan de Fuca Ridge is an intermediate spreading center which is magmatically more robust and highly influenced by the presence of the nearby hotspots. Moreover, our study is focused on much younger lithosphere (<2.67 My old), compared to 40-70 My old lithosphere that was focus of the study in the Equatorial Atlantic. This means that the top of the LAB in our study is expected at much shallower depths (~11 km below seafloor) compared to the old crust (72-88 km below seafloor).

In fact, we are pleasantly surprised to image 2-3 reflections between 11 and 14 km below the seafloor, which makes our study important and exciting. As we apply these techniques to different locations, we are likely to discover new facet of the lithosphere and the LAB.

The modeling in Fig. S4 should be more quantitative in terms of comparison to data, and should be discussed much more in the main text. As is the result presented in the main text is not quantitative in terms of what structure is required by the data.

We agree with Reviewer and have performed more modelling, and some of the key results are brought into the main text, along with one figure (i.e., new Figure 4).

Line 152 – 152 – No petrological mechanism exists to give a high velocity layer? What about frozen-in ultra mafics? I also found it curious that there was no reference to [Ohira et al., 2017 Evidence for frozen melts in the mid-lithosphere detected from active-source seismic data (Scientific Reports)]. In this paper a very similar observation was interpreted as a frozen-in MLD.

We agree with Reviewer, and have: (1) removed the above statement, (2) performed modelling to explain that thin high velocity would not be able to produce required synthetic seismograms, (3) included Ohira et al (2017) in the discussion, and (4) suggest that frozen melt sills could be one of explanation for our observation and discuss the limits of the two possible interpretations.

It is good that the authors compared to the receiver functions, but there has been a lot of other excellent work in this particular region of Cascadia (by Bell & Forsyth; Toomey; Gao; Audet; and many others), but references and comparisons to these – where appropriate - were lacking. The result could have been put into context better.

We have cited some of the suggested papers and have discussed relationship with our results. For example, the results of Bell and Forsyth starts from 25 km depth, much below our images. We have also included some of these results in the discussion section.

Line 170: The authors state that the lithosphere thickens along the 1200 degree isotherm in the plate cooling model. This is not true/proscribed by that model.

The LAB at 1200°C isotherm is an assumption, especially when we assume that melt is present at the base of the lithosphere. Solidus for dry basalt is about 1100°C (e.g., Lamber and Willie, 1972) near the surface, would be about 1200°C at 11-14.5 km depth.

We have discussed this point in the text to make our point.

The reasons for the flatness of the interface are presented in a confusing way, without enough logic, support detail.

We have now rewritten this section to explain different arguments about the flatness of the reflections.

Line 173 – 176: To explain the flatness of the reflector, the authors call on hydrothermal circulation, then state that it would not be enough, or only help to 0.3 My with a reference to Roger Buck personal communication (?!). This is far too casual. More logic, explanation, background is required. Include calculations with references would be ideal. Also, why is hydrothermal needed? At intermediate spreading centers a thickened plate is predicted beneath the ridge. More logic is needed.

We have included a modelling from Roger Buck and have invited him to be a co-author to acknowledge his contribution. In addition, we have also included modeling results in Fig. 6B. We have provided the logic required explain the flatness of the LAB and have also given other possible models that may explain the observation.

Then the authors call on a sediment blanket to increase heat at older ages caused by increased terrestrial sediments. However, most of the terrestrial sediment gets stuck on the other side of the trench, so not clear this is true. Estimates for sediment thickness near the ridge are typically < 1 km. More logic/support/detail would help.

Although most of the terrestrial sediments get stuck near the trench, some sediments do reach the spreading centre and young oceanic lithosphere as indicated by seismic image and borehole data. The presence of sediment acts as a blanket and hence keeps the lithosphere warmer, as suggested by the images of shallow magma chambers at sedimented Andaman Sea spreading centre (e.g. Jourdan et al, Geology, 2016). A recent study from the Atlantic Ocean (Audhkhasi and Singh, GGG, under final stage of acceptance, see the figure blow) shows that the thinning of layer 2A is inversely related to the thickening of the sediments, which happens in the first 4 Myr, indicating rapid decrease in active hydrothermal circulation with age.

Figure from Audhkhasi and Singh (GGG, under final stage of acceptance): Layer 2A and sediment thicknesses as a function age of the crust. The Layer 2A thickness is inversely related to the sediment thickness for the first 4 Myr, indicating a special thermal regime for the first four Myr.

The heat flow study and drilling results indicate that the basement temperature in our study area could be up to 60° C (instead 0-2° C) because of this blanketing effect. We have added more text from the heat flow study and drilling results to make the point.

Line 209: the mechanism for flattening described in this section is hotspots, but without a clear link to the previous interpretation (sediment). More information about what is actually required by the data would help, or even more connection between the interpretations.

We have added more explanation to clarify the point.

Line 211 – 212 not sure how hotspot anomalies get sheared or dragged?

We have removed this statement as it was rather confusing.

Line 230 onwards has a lot of strong statements about implications, but most are too brief/terse. All require a few more sentences of logic to connect.

We have rewritten this section and added more explanations.

Minor comments:

Line 33, 40, and 2017: these statements are vague and overly negative to surface waves and receiver functions. Surface waves lateral resolution is not always 100s km – it is smaller in regional experiments and at shallow depths, so that statement is incorrect. There is a reference to an Eaton review paper, but not clear what Eaton was talking about, maybe global studies of much deeper depths (?). It would probably better to go with a surface

wave paper that does testing and make the statement more specific. Also, receiver function depth resolution is typically better than 10 – 15 km at lithospheric depths, so again incorrect, and the statement would need a reference to back it up.

As it is rather difficult to find papers that could really provide theoretical quantitative uncertainty on different methods, we have deleted the above statements and have included precise information from Bell et al. (2016) and Rychert et al (2018), both studies from the same region, highlight the limits of the surface and receiver function method in the present context.

Line 38 – 39 Are the receiver function depths below sea surface or seafloor? This is Important for comparison.

The depths are given below sea surface.

Lines 213 – 215 need to be referenced.

This statement has been removed in the revised version of the manuscript.

Line 208- 209 this is a vague comparison. It should be more specific and quantitative.

This point has been modified to include observations from Bell et al. (2016) and Rychert et al. (2018).

Reviewer #3 (Remarks to the Author):

Comments on the manuscript “Thin Lithosphere and Mantle Thermal Anomaly beneath the Young Juan de Fuca Plate” by Yanfang Qin et al.

I was left confused by the interpretation presented in this manuscript. Maybe it is a problem of organization, but there appear to be conflicting statements and assumptions being made that left me questioning the validity of the conclusions. The authors should address or clarify the following:

Using wide-angle data to generate a pre-stack depth migrated (PSDM) reflection image is an interesting idea. But this method deserves a more thorough analysis than is presented here, specifically to demonstrate the strengths and pitfalls of the approach. The synthetics presented in Fig S4 are a promising start. At the least, the authors should provide some examples of what happens when the synthetic seismograms shown in Figure S4 are processed using their PSDM approach. How does an imperfect assumed velocity model distort the reflection image? Etc.

We appreciate the reviewers point and have carried out PSDM on synthetic data with different velocities to address the above point, and have included in the Supplementary Information. We

have also brought the PSDM and modeling work in the main part of the text and hope this will address the points raised by the reviewer.

L109. An age and hence temperature dependent mantle velocity is applied in the PSDM, where velocity increases with plate age. Does this not conflict with later argument that the mantle isotherms are flat? The relative velocity differences are small and may not change the imaging results much, but this should be corrected or justified.

The variation of mantle velocity with age is small (7.65 to 7.8 km/s) in the study area, and does not affect imaging results significantly (new Figure S9). Therefore, we have used a constant velocity for migration to make sure that all the results are consistent.

L185. This paragraph appears to contradict the preceding paragraph and makes an unphysical claim. The authors first invoke vigorous hydrothermal circulation in order to rapidly depress the mantle isotherms and flatten them away from the axis for a short distance (~0.3 Myr equivalent). Then, the authors refer to drilling results that support much warmer basement temperatures relative to unsedimented young seafloor. If significant heat is already removed near the ridge axis then adding sediment and stopping the advective heat loss may warm the basement temperature, but it does not increase mantle temperatures. It simply slows down the rate of mantle temperature decay with age. I think the authors intended to say that in addition to sedimentation cutting off hydrothermal circulation, an unrelated heat source is required that flattens the 1200 degC isotherm by offsetting conductive heat loss as the plate ages. However, ref. 37 directly states that the heat flow on the sedimented JdF.

As shown in Figure 6B, the active hydrothermal circulation would make the isotherm steeper near the ridge axis, and flatten the 800°C isotherm, even raise the lower temperature isotherms, but would not be able to flatten the 1200°C. Adding of the sediments would keep the mantle warmer, but we agree with the reviewer that we would need a thermal anomaly to bring the 1200°C isotherm to a shallower depth and make it flatter. We have rewritten the discussion.

Plate in the Endeavor segment does not match predictions from plate cooling models. Flattening the 1200°C isotherm to the point where temperatures are warmer than plate cooling predictions should yield anomalously high heat flow, which is not the case here. Not only that, the heat flow would be constant with age to reflect a flat isotherm (e.g. no change in temperature with depth), again this is not the case. Sediment cover merely limits cooling to conductive heat loss.

All the observations (receiver function, tomography and our results) indicate that the base of the lithosphere is flat and does not follow the plate-cooling model. The heat flow data have a lot of scatter but the over all heat flow is low over 50 km from the ridge axis (Davis et al). The only point we wish to make is that there would be an excess of temperature due to the presence of sediments.

L199. The authors state that the subsidence analysis suggests 30°C warmer mantle. Is this enough to sufficiently shallow the isotherms between 0.5 and 2.7 Myr? Intuitively seems unlikely, and does not appear to be the case based on isotherm contours in Fig 3 as well. How would warmer background mantle manifest as flat isotherms? Furthermore, the subsidence analysis does not consider the effect of crustal thickness variations on basement

depth, which is critical when inferring warmer mantle (based on isostasy, thicker crust equals shallower seafloor and hence no thermal anomaly is needed). The gravity anomalies in the region cannot be ignored either.

We agree with the reviewer that 30°C is not sufficient to shallow the isotherms entirely. The presence of the sediments represents one of the possible factors that could contribute to shallower isotherms. To explain the shallow isotherms we still need the existence of the mantle thermal anomaly.

L224. Can the authors distinguish between melt pockets and frozen sills? Is it not simpler to interpret the reflectors as frozen sills and thereby avoid the complex and seemingly inconsistent thermal interpretation required by melt?

As stated in the revised text, we cannot distinguish between molten melt sill and frozen melt sill, and we state that it is possible that these reflections could be due to frozen melt sills. We have taken the reviewers advice and explained both models, their pros and cons.

L229. Again, sediment blanketing does not increase mantle temperature, heat is still lost via conduction. Indeed, plate cooling models ignore hydrothermal circulation and only consider conductive heat loss that is representative of sedimented oceanic plates. This leaves a 30°C mantle temperature increase inferred from the subsidence analysis in Fig S8, which as previously noted is problematic in itself and requires additional justification.

To shallow isotherms we do need the existence of mantle thermal anomaly the presence of which can be inferred by the existence of several seamount chains on the west flank, the skewed mantle suggested by upper mantle velocity study (Arnoux et al., 2019) and 40 km wide plateau centered at the ridge axis (Carbotte et al., 2008).

Minor points:

L122. This statement is confusing. Why does the downward continued data, which is muted, yield reverberations?

Pg arrivals have been muted but not the rear offset data, which create reverberation when migrated.

L136. What exactly are the uncertainties in mantle velocity structure? This relates to a previous point regarding L109.

We do the tests using end-member velocity models by varying upper mantle velocities by +/- 500 km/s. The results are shown in new Figure S6.

L155. What is the minimum Vp reduction required to generate the reflections? Is it 8.5%?

As the modelling and AVO studies indicate, the variation should be 7-8% decrease in velocity, which has been made clear.

L200. Authors should explicitly say that they are using the subsidence analysis described in ref. 38 to avoid confusion.

We modify the sentence as follows to address Reviewer's comment.

Reviewers' comments:

Reviewer #2 (Remarks to the Author):

The result is still interesting and important. The authors have cleared up some things for instance- depths to the discontinuities are now presented consistently. However, the manuscript still needs work to improve clarity. Now there is a mix of melt sill and frozen-gabbro interpretations. This is OK, but while the abstract is clear, the ideas are not clearly presented throughout the entire manuscript. The discussion needs work to clearly explain the possibilities and to put the results in context. The concluding remarks need to be much more clearly presented. Some very basic things from my previous review were not addressed such as accounting for lateral heat conduction and putting the results in the context of previous work.

major

Overall, the discussion section still needs work.

231- 236 is a repeat of the introduction. Authors should take the tone of a discussion, putting their result in context. Although the authors explained why they can see lenses here, whereas M&S saw the top and the bottom of the LAB in the response to review, the explanation/comparison is lacking in the discussion.

Line 199 - 200 -7% decrease in velocity could be associated with the presence of a frozen gabbroic sill while - Need a reference and more specifics. What is the background composition.

Line 245: The authors have improved the discussion of LAB or not LAB, etc. But, there are still parts that are strange, and things that I brought up in my previous review that are ignored. "If we assume that the base of the lithosphere corresponds to the 1200° C isotherm in the plate-cooling model, then the lithosphere should thicken from ~3 km below the ridge axis to about 16 km at 2.65 Myr." Where do these numbers come from? In the plate cooling model the lithosphere is 0 km thick by definition at the ridge axis. In another section (line 281) the authors talk about thick sediment moving the 1200deg isotherm shallower, which is not clearly described well given that it would move 1200 deg. to negative depths in the classic plate model of Stein & Stein & others. Also, hydrothermal circulation is important. But as I mentioned in my last review - thick lithosphere is predicted beneath the ridge axis in geodynamic models that account for lateral heat conduction (Morgan, et al., 1987). Although this is not addressed. This should be addressed first, and then add the hydrothermal circulation.

line 343: The concluding remarks should be more clear. The authors seem to go back and forth on melt sills and frozen-in features, in the end saying they would need a different experiment to tell the difference. But the logic is confusing. Surface waves indicate no thermal anomaly so there could be frozen sills, but another surface wave model shows fast velocities to 28 km depth so no evidence of axial chamber to produce the sill? How could no thermal anomaly support frozen-in sills and fast velocities not support sills?

Presenting results with a \sim is not very strong. Error bars are better.

The response on anisotropy should be better explained. Couldn't there be a shear zone?

Minor

I am confused by the authors' lack of reference to several papers including especially Audet et al., 2016 GJI, Receiver functions using OBS data: promises and limitations from numerical modelling and examples from the Cascadia Initiative. This study found a discontinuity further south on the ridge axis and at greater depth. Does the difference in depth mean that ridge processes vary along axis? Or is there a way that the receiver function study would not see discontinuities at the depths proposed by the authors.

Line 304: "The S-wave receiver function results show the presence of a sub-horizontal converted S-wave receiver function between 20 and 45 km depth¹² beneath the JdF plate" What does depth beneath the JdF plate mean, that would be much deeper in the asthenosphere? It should actually be depth beneath the sea surface.

Line 39 to 44: "A receiver function study over 0 to 8 Myr old lithosphere on the Juan de Fuca (JdF) Plate showed the presence of a phase converted S-wave sub-horizontal interface between 20 and 45 km beneath sea surface¹², suggesting that receiver function methods would also have some difficulty in precisely deciphering the depth of the LAB for a young oceanic lithosphere." This statement does not make sense. This phrase should be deleted: "suggesting that receiver function methods would also have some difficulty in precisely deciphering the depth of the LAB for a young oceanic lithosphere."

Line 54 remove "very first" claims of first are almost never correct and always offensive to somebody.

There are some grammatical errors.

Morgan, J. P., Parmentier, E. M., and Lin, J. (1987), Mechanisms for the origin of midocean ridge axial topography—implications for the thermal and mechanical structure of accreting plate boundaries, *J. Geophys. Res.*, 92(B12), s12823–12836. doi:10.1029/JB092iB12p12823.

Reviewer #3 (Remarks to the Author):

Comments on the revised manuscript “Thin versus Thick Lithosphere beneath the Young Juan de Fuca Plate” by Yanfang Qin et al.

The authors have addressed most of my concerns regarding the original manuscript. The additional technical tests make a stronger case for the imaging results, and the overall interpretation is more robust. This is an interesting result with important implications for lithosphere heterogeneity. However, the updated interpretation contradicts itself and is incoherent in some sections of the manuscript. Clarifications are needed because of this, which should be relatively straightforward for the authors to implement.

L33. Hydrothermal circulation will cool the plate, which increases its thickness. Replace “result in changing the thickness” to “result in increasing the thickness”.

L42. Why does an interface depth between 20 and 45 km depth suggest that the receiver function method is having difficulty constraining the LAB depth? Should elaborate or discard this claim.

L50. This is a good place to also mention that MT results have observed a thin high conductivity channel interpreted to be a partial melt channel, consistent with the active source seismic data (currently ref. 47, Naif et al).

L110. "As synthetic.." should be "A synthetic.."

L160. As a side note, I would argue that the mantle reflectors are much more prominent in the southern half of the profiles, and relatively weak and intermittent in the northern half. Curious if the authors think this may be meaningful.

L200. Why is a -7% velocity reduction attributed to a frozen gabbro sill and a -30% reduction partial melt sill? This is not clear, particularly since the velocity reductions interpreted as partial melt are ~-8% in refs. 13 and 14. It may be best to exclude linking the magnitude of the reduction to a particular interpretation, unless the authors can provide supporting evidence.

L204-208. From what I can tell, it is the observation of a negative polarity reflection in the real data that supports a low velocity layer, not the modeling study. In fact, the authors specifically note that the model of a 300m layer with +7% velocity increase also resembles the data (L192). This section should be clarified to only state that the authors prefer low velocity interpretation because of the noted negative polarity, and refrain from suggesting that the modeling favors low velocity features.

L250. Add reference to Methods section for description of the thermal modeling.

L261. Making the claim that a ridge-perpendicular profile is needed to differentiate between partial melt and frozen sills implies that there is a way to distinguish between the two with this type of data. As I alluded to in the comment on L200, it is not clear to me that frozen versus partial melt sills possess unique velocities or other attributes. If so then provide supporting evidence for this claim.

L264-270. Both of these scenarios seem comparable. Both need a flattened 1200 degC isotherm, the only difference being the depth to the isotherm. This makes the statement on L273 problematic, since if the second scenario is less likely, then so is the third scenario. If the authors want to stick with the claim that only the second scenario is less likely of the two, then an explanation is needed.

L283. Although I get what the authors mean to say, this sentence conflates two processes and should be removed. Adding more sediment has little effect relative to advective heat loss by removing most sediment. It is more plausible to attribute the 60 degC warming to the fact that non-sedimented seafloor experiences addition heat loss through advection, not because adding sediment makes the plate warmer relative to plate cooling models. This also distracts from the more valuable discussion beginning on L285.

L304-318. This paragraph is hard to follow. The authors claim that the receiver function results in ref. 12 suggest flat lithosphere thickness, yet note 20-45 km depths (hardly seems flat). They then argue that this contradicts surface wave tomography results in ref. 8, which found linearly increasing lithosphere thickness with age from 25 km to 65 km. While the two references may not be in good agreement, this may only be the case when comparing the entire JdF plate system as a whole. The wide-angle reflection profiles presented here are isolated to a confined region specific to the Endeavor ridge segment. It is hard to compare with the figures presented in refs. 8 and 12 since they only show cross-sections of their observations for the Axial ridge segment and not the Endeavor segment. If the authors' imperative is to support the claim that the lithosphere is thick and potentially colder, then simply state so and cite both refs. 8 and 12. Trying to argue that they are inconsistent with one another is counterproductive and distracting.

L343. But doesn't this then conflict with all of L285-300? The conclusion section completely sidesteps the evidence for a warm thermal anomaly presented earlier in the manuscript. It is critical for these two conflicting arguments to be reconciled.

L346-350. If ref 45 is inconsistent with melt lenses at the ridge axis, then so are refs 8 and 12. They all show evidence for a ~25 km thick lithosphere near the ridge. It makes more sense to simply point out that the tomography and receiver functions may lack the resolution. We know there is melt at the ridge axis regardless of the thermal models since it has been observed. The authors can also cite Eilon & Abers (2017) to support their argument for a relatively narrow melt zone in Cascadia.

Eilon, Z. C., & Abers, G. A. (2017). High seismic attenuation at a mid-ocean ridge reveals the distribution of deep melt. *Science Advances*.

L355. See comment on L261.

Final comment (more of a suggestion). The results and interpretation imply frozen melt sills are potentially widespread in oceanic lithosphere. One might then deduce that such sills may be discovered in existing data sets in other oceans. Given the implication of such sills on lithosphere heterogeneity, I would argue this is a more appropriate point to end on than the dip of the subducting plate, which is a thermal buoyancy related issue. Although both are important.

Please note the reviewers comments in blue and response in black

Reviewer #2 (Remarks to the Author):

The result is still interesting and important. The authors have cleared up some things for instance- depths to the discontinuities are now presented consistently. However, the manuscript still needs work to improve clarity. Now there is a mix of melt sill and frozen-gabbro interpretations. This is OK, but while the abstract is clear, the ideas are not clearly presented throughout the entire manuscript. The discussion needs work to clearly explain the possibilities and to put the results in context. The concluding remarks need to be much more clearly presented. Some very basic things from my previous review were not addressed such as accounting for lateral heat conduction and putting the results in the context of previous work.

major

Overall, the discussion section still needs work.

231- 236 is a repeat of the introduction. Authors should take the tone of a discussion, putting their result in context. Although the authors explained why they can see lenses here, whereas M&S saw the top and the bottom of the LAB in the response to review, the explanation/comparison is lacking in the discussion.

We agree with the reviewer and have re-written this section to take these points into account, especially toned as a discussion.

Line 199 - 200 **-7% decrease in velocity could be associated with the presence of a frozen gabbroic sill while** - Need a reference and more specifics. What is the background composition.

We have provided a reference. The background rock is peridotite and has been stated clearly.

Line 245: The authors have improved the discussion of LAB or not LAB, etc. But, there are still parts that are strange, and things that I brought up in my previous review that are ignored. "If we assume that the base of the lithosphere corresponds to the 1200° C isotherm in the plate-cooling model, then the lithosphere should thicken from ~3 km below the ridge axis to about 16 km at 2.65 Myr." Where do these numbers come from? In the plate cooling model the lithosphere is 0 km thick by definition at the ridge axis.

The 3 km depth comes from the depth of the axial melt lens (Carbotte et al., 2008), even though the plate cooling model assumes 0 km thickness for the lithosphere at the ridge. We have addressed this point along with the point below in the revised paragraph.

In another section (line 281) the authors talk about thick sediment moving the 1200deg isotherm shallower, which is not clearly described well given that it would move 1200 deg. to negative depths in the classic plate model of Stein & Stein & others. Also, hydrothermal circulation is important. But as I mentioned in my last review - thick lithosphere is predicted beneath the ridge axis in geodynamic models that account for lateral heat conduction (Morgan, et al., 1987). Although this is not addressed. This should be addressed first, and then add the

hydrothermal circulation.

We have brought in the discussion from Morgan et al. (1987) and rewritten this section as suggested by the reviewer.

line 343: The concluding remarks should be more clear. The authors seem to go back and forth on melt sills and frozen-in features, in the end saying they would need a different experiment to tell the difference. But the logic is confusing. Surface waves indicate no thermal anomaly so there could be frozen sills, but another surface wave model shows fast velocities to 28 km depth so no evidence of axial chamber to produce the sill? How could no thermal anomaly support frozen-in sills and fast velocities not support sills?

We have sharpen and shorten this section, and made the message clear.

Presenting results with a ~ is not very strong. Error bars are better.

It would be really hard to have error bars well constrained given the data we are working with. There are many uncertainties associated in these estimations so giving an arbitrary error bars would not be judicial.

The response on anisotropy should be better explained.

In the last review, the reviewers has mentioned ‘What about anisotropy? Could this influence and/or explain your result? Many papers have attributed apparent lithospheric discontinuities to anisotropy (including for example, Audet et al., 2016 – at the Juan de Fuca Ridge)’ we had responded saying ‘Our results show the presence of wide-angle reflections, which require sharp velocity contrasts (vertically). Though anisotropy might be present on larges-scale, on short wavelength scale one would require lithological variations, e.g. frozen melt, dunite, or molten melt etc.’

Audet et al (2016) find an unrealistically high anisotropy of 26-30% in 8-15 km thick layer below the Moho. Such a high anisotropy would lead to a P-wave velocity of up to 10.6 km/s below the Moho. Hundreds of refraction experiments have been performed in oceanic lithosphere, and never such a high velocity have been observed, and therefore, we do not believe that such a high anisotropy exist in the mantle below the Moho. A P-wave anisotropy of 3-7% have been observed in the Pacific (Armaux et al., 2016). A 7% anisotropy (P-wave velocity from 8 to 8.5 km/s) could produce the required +7% equivalent reflections, but this anisotropic layer has to be thin as suggested by modeling, and therefore the conclusion would be similar.

Couldn't there be a shear zone?

Dziak (2006, Geology) propose the idea of large shear zones in the area. Most of the deformation in the area are associated with strike-slip motion, creating nearly vertical faults. However, our reflectors are sub-horizontal and extend over 60-70 distance from the ridge axis, which cannot the generated from strike-slip type of deformation. One can produce ridge-perpendicular deformation at the base of the lithosphere due to extensional processes, but we do not observe any earthquakes at 11-15 km depth below the seafloor at intermediate spreading centres, and hence they cannot be due to shearing. Therefore, we have not mentioned shearing as a possible explanation.

Minor

I am confused by the authors' lack of reference to several papers including especially Audet et al., 2016 GJI, Receiver functions using OBS data: promises and limitations from numerical modelling and examples from the Cascadia Initiative. This study found a discontinuity further south on the ridge axis and at greater depth. Does the difference in depth mean that ridge processes vary along axis? Or is there a way that the receiver function study would not see discontinuities at the depths proposed by the authors.

We are aware of this paper, and have now discussed in the paper. In order explain the presence of 25-30% of anisotropy in the mantle below the Moho, one would require a P-wave velocity of up to 10.25-10.66 km/s below the Moho down to 8-15 km in the mantle, which is unrealistically high. No active source data have shown such a high velocity below the Moho. However, we agree with the negative amplitude signal at 2 s, requiring a low velocity underneath, which could possibly be the base of the lithosphere. We have added these points in the paper.

Line 304: "The S-wave receiver function results show the presence of a sub-horizontal converted S-wave receiver function between 20 and 45 km depth¹² beneath the JdF plate" What does depth beneath the JdF plate mean, that would be much deeper in the asthenosphere? It should actually be depth beneath the sea surface.

We agree with the reviewer and clarified this point.

Line 39 to 44: "A receiver function study over 0 to 8 Myr old lithosphere on the Juan de Fuca (JdF) Plate showed the presence of a phase converted S-wave sub-horizontal interface between 20 and 45 km beneath sea surface¹², suggesting that receiver function methods would also have some difficulty in precisely deciphering the depth of the LAB for a young oceanic lithosphere." This statement does not make sense. This phrase should be deleted: "suggesting that receiver function methods would also have some difficulty in precisely deciphering the depth of the LAB for a young oceanic lithosphere."

We have rewritten this sentence to make the point more clear.

Line 54 remove "very first" claims of first are almost never correct and always offensive to somebody.

Removed

There are some grammatical errors.

We have tried out best to address this point.

Reviewer #3 (Remarks to the Author):

Comments on the revised manuscript "Thin versus Thick Lithosphere beneath the Young Juan de Fuca Plate" by Yanfang Qin et al.

The authors have addressed most of my concerns regarding the original manuscript. The

additional technical tests make a stronger case for the imaging results, and the overall interpretation is more robust. This is an interesting result with important implications for lithosphere heterogeneity. However, the updated interpretation contradicts itself and is incoherent in some sections of the manuscript. Clarifications are needed because of this, which should be relatively straightforward for the authors to implement.

L33. Hydrothermal circulation will cool the plate, which increases its thickness. Replace “result in changing the thickness” to “result in increasing the thickness”.

We have rewritten this sentence taking the reviewers 2 point into account.

L42. Why does an interface depth between 20 and 45 km depth suggest that the receiver function method is having difficulty constraining the LAB depth? Should elaborate or discard this claim.

We have rewritten this section to elaborate out point.

L50. This is a good place to also mention that MT results have observed a thin high conductivity channel interpreted to be a partial melt channel, consistent with the active source seismic data (currently ref. 47, Naif et al).

We have added a sentence to address this point.

L110. “As synthetic..” should be “A synthetic..”
The text is corrected accordingly.

L160. As a side note, I would argue that the mantle reflectors are much more prominent in the southern half of the profiles, and relatively weak and intermittent in the northern half. Curious if the authors think this may be meaningful.

The problem is that for the youngest age, we have one OBH missing in the north, which produces smeared anomaly that seems to be stronger than the one observed in the south. Also looking at Figure S5, we do not see the pattern at all. This means that 2 profiles out of 4 show the pattern suggested by the reviewer, and given this is a minor point we have not gone into depth.

L200. Why is a -7% velocity reduction attributed to a frozen gabbro sill and a -30% reduction partial melt sill? This is not clear, particularly since the velocity reductions interpreted as partial melt are ~-8% in refs. 13 and 14. It may be best to exclude linking the magnitude of the reduction to a particular interpretation, unless the authors can provide supporting evidence.

We have elaborated this point and have explained the difference between frozen sills, partially molten sills and large amount of melt zone in the text

L204-208. From what I can tell, it is the observation of a negative polarity reflection in the real data that supports a low velocity layer, not the modeling study. In fact, the authors specifically note that the model of a 300m layer with +7% velocity increase also resembles the data (L192). This section should be clarified to only state that the authors prefer low

velocity interpretation because of the noted negative polarity, and refrain from suggesting that the modeling favors low velocity features.

We have rephrased this sentence to take the reviewers point into account.

L250. Add reference to Methods section for description of the thermal modeling.

We have rewritten this section and have referenced the method section.

L261. Making the claim that a ridge-perpendicular profile is needed to differentiate between partial melt and frozen sills implies that there is a way to distinguish between the two with this type of data. As I alluded to in the comment on L200, it is not clear to me that frozen versus partial melt sills possess unique velocities or other attributes. If so then provide supporting evidence for this claim.

We have removed this sentence.

L264-270. Both of these scenarios seem comparable. Both need a flattened 1200 degC isotherm, the only difference being the depth to the isotherm. This makes the statement on L273 problematic, since if the second scenario is less likely, then so is the third scenario. If the authors want to stick with the claim that only the second scenario is less likely of the two, then an explanation is needed.

In order to decrease confusion, and take physics into account, we have decreased the possible models to two, and elaborate in these two models.

L283. Although I get what the authors mean to say, this sentence conflates two processes and should be removed. Adding more sediment has little effect relative to advective heat loss by removing most sediment. It is more plausible to attribute the 60 degC warming to the fact that non-sedimented seafloor experiences addition heat loss through advection, not because adding sediment makes the plate warmer relative to plate cooling models. This also distracts from the more valuable discussion beginning on L285.

We have re-organized the text, and slightly modified this sentence.

L304-318. This paragraph is hard to follow. The authors claim that the receiver function results in ref. 12 suggest flat lithosphere thickness, **yet note 20-45 km depths (hardly seems flat)**. They then argue that this contradicts surface wave tomography results in ref. 8, which found linearly increasing lithosphere thickness with age from 25 km to 65 km. While the two references may not be in good agreement, this may only be the case when comparing the entire JdF plate system as a whole. The wide-angle reflection profiles presented here are isolated to a confined region specific to the Endeavor ridge segment. It is hard to compare with the figures presented in refs. 8 and 12 since they only show cross-sections of their observations for the Axial ridge segment and not the Endeavor segment. If the authors' imperative is to support the claim that the lithosphere is thick and potentially colder, then simply state so and cite both refs. 8 and 12. Trying to argue that they are inconsistent with one another is counterproductive and distracting.

We agree with the reviewer and modified the text to clarify these points.

L343. But doesn't this then conflict with all of L285-300? The conclusion section completely sidesteps the evidence for a warm thermal anomaly presented earlier in the manuscript. It is critical for these two conflicting arguments to be reconciled.

We have added paragraph to bring out the idea of the thermal anomaly.

L346-350. If ref 45 is inconsistent with melt lenses at the ridge axis, then so are refs 8 and 12. They all show evidence for a ~25 km thick lithosphere near the ridge. It makes more sense to simply point out that the tomography and receiver functions may lack the resolution. We know there is melt at the ridge axis regardless of the thermal models since it has been observed. The authors can also cite Eilon & Abers (2017) to support their argument for a relatively narrow melt zone in Cascadia.

Eilon, Z. C., & Abers, G. A. (2017). High seismic attenuation at a mid-ocean ridge reveals the distribution of deep melt. *Science Advances*.

We have cited this paper and mentioned about the resolution issue.

L355. See comment on L261.

This sentence has been deleted.

Final comment (more of a suggestion). The results and interpretation imply frozen melt sills are potentially widespread in oceanic lithosphere. One might then deduce that such sills may be discovered in existing data sets in other oceans. Given the implication of such sills on lithosphere heterogeneity, I would argue this is a more appropriate point to end on than the dip of the subducting plate, which is a thermal buoyancy related issue. Although both are important.

We have added a sentence.

REVIEWERS' COMMENTS:

Reviewer #3 (Remarks to the Author):

Comments on the re-revised manuscript "Thin versus Thick Lithosphere beneath the Young Juan de Fuca Plate" by Yanfang Qin et al.

The authors have addressed my concerns regarding the revised manuscript. I have a few simple corrections related to some references.

L48. This seems to be a misreading of ref 13. The 26-30% anisotropy is not relative to the mean, but to the fast/slow axes. It is more like 8.2 ± 1.2 km/s. See their Figure 13b.

L51. The receiver function study in ref 14 specifically notes an error of ± 5 km in their depth resolution, not ~ 20 km as implied here. It is more accurate to state that the receiver functions may lack the resolution to image thin sills less than a few km thick.

L408. Ref 55 argues for a narrow upwelling zone that suggests dynamic upwelling at ridges. It does not necessarily support high attenuation extending to 4 Ma. Their Fig 3 shows differential attenuation relative to 4-8 Ma reference value. It is not clear to me that ref 55 supports a warm thermal anomaly.

REVIEWERS' COMMENTS:

Reviewer #3 (Remarks to the Author):

Comments on the re-revised manuscript "Thin versus Thick Lithosphere beneath the Young Juan de Fuca Plate" by Yanfang Qin et al.

The authors have addressed my concerns regarding the revised manuscript. I have a few simple corrections related to some references.

L48. This seems to be a misreading of ref 13. The 26-30% anisotropy is not relative to the mean, but to the fast/slow axes. It is more like 8.2 ± 1.2 km/s. See their Figure 13b.

We have change this to 8.2 ± 1.2 km/s as suggested by the reviewer.

L51. The receiver function study in ref 14 specifically notes an error of ± 5 km in their depth resolution, not ~ 20 km as implied here. It is more accurate to state that the receiver functions may lack the resolution to image thin sills less than a few km thick.

Although we are aware that ± 5 km resolution has been stated for receiver function studies, half a wavelength ($\lambda/2$) (half period) is 20 km as stated in the paper based on the results. Assuming that resolution is $\lambda/4$, the resolution in the present circumstances will be ± 10 km, not ± 5 km. In fact, to be on the safe side, we generally say that resolution is between $\lambda/2$ and $\lambda/4$ (Vireux and Operto, 2009). We have revised the text to address this point.

L408. Ref 55 argues for a narrow upwelling zone that suggests dynamic upwelling at ridges. It does not necessarily support high attenuation extending to 4 Ma. Their Fig 3 shows differential attenuation relative to 4-8 Ma reference value. It is not clear to me that ref 55 supports a warm thermal anomaly.

We have modified this text, and replaced 3-4 Myr with 'near the ridge axis'